# Trogocytic molting of T cell microvilli upregulates T cell receptor surface expression and promotes clonal expansion

Jeong-Su Park[1,2], Jun-Hyeong Kim [1,2], Won-Chang Soh[1,2], Na-Young Kim[1,2], Kyung-Sik Lee[1,2], Chang-Hyun Kim[1,2], Ik-Joo Chung[3], Sunjae Lee[1], Hye-Ran Kim [1,2,4] ✉ & Chang-Duk Jun [1,2] ✉

Although T cell activation is known to involve the internalization of the T cell antigen receptor (TCR), much less is known regarding the release of TCRs following T cell interaction with cognate antigen-presenting cells. In this study, we examine the physiological mechanisms underlying TCR release following T cell activation. We show that T cell activation results in the shedding of TCRs in T cell microvilli, which involves a combined process of trogocytosis and enzymatic vesiculation, leading to the loss of membrane TCRs and microvilli-associated proteins and lipids. Surprisingly, unlike TCR internalization, this event results in the rapid upregulation of surface TCR expression and metabolic reprogramming of cholesterol and fatty acid synthesis to support cell division and survival. These results demonstrate that TCRs are lost through trogocytic 'molting' following T cell activation and highlight this mechanism as an important regulator of clonal expansion.

Numerous studies over the past three decades have aimed to understand the molecular mechanisms through which the expression of the T cell receptor (TCR) is regulated on the surface of resting and activated T cells. It is widely accepted that TCR downmodulation occurs due to TCR internalization, decreased recycling, and increased degradation[1–6]. However, relatively few studies have reported that the TCR is released from activated T cells in the form of extracellular vesicles (EVs), including exosomes, microvesicles, or lytic granules[7,8]. Blanchard and colleagues demonstrated that TCR triggering induces exosomes bearing the TCR/CD3/ζ complex in human T cells[9]. Dustin et al. demonstrated that T cells produce TCR-enriched microvesicles or synaptic ectosomes at the center of the immunological synapse[10,11]. Independent of previously published works, we and others have identified that TCRs are highly enriched in microvilli tips[12,13]. Surprisingly, we observed that microvilli were fragmented into TCR-enriched, nanosized membrane particles (T cell microvilli particles, or TMPs) due to the combined action of two independent mechanisms: trogocytosis

and enzymatic vesiculation. Released microvesicles or membrane particles trigger signaling in antigen-bearing B cells[10] and dendritic cells (DCs)[13], indicating that these nanosized membrane particles can transfer T cell-derived signals to their cognate APCs. Therefore, TMPs can be regarded as T cell immunological synaptosomes (TISs)[13]. Interestingly, unlike exosomes, which are secreted from multivesicular bodies, the release of surface microvilli covering the cell outer layer is reminiscent of molting. In the present study, because T cells also shed their surface microvilli in a combined process of trogocytosis and enzymatic breakage, we termed this phenomenon trogocytic 'molting'. At the cellular level, including in mammalian immune cells, it remains unclear whether this molting of T cell microvilli is associated with their function or proliferation.

Although it is thought that TCR downregulation occurs mainly as the result of internalization and degradation, we demonstrate in this study that trogocytic microvilli shedding is a common process resulting in the loss TCR expression following interaction with cognate

[1]School of Life Sciences, Gwangju Institute of Science and Technology (GIST), Gwangju 61005, Republic of Korea. [2]Immune Synapse and Cell Therapy Research Center, Gwangju Institute of Science and Technology (GIST), Gwangju 61005, Republic of Korea. [3]Department of Hematology-Oncology, Immunotherapy Innovation Center, Chonnam National University Medical School, Hwasun 58128, Republic of Korea. [4]Division of Rare and Refractory Cancer, Tumor Immunology, Research Institute, National Cancer Center, Goyang 10408, Republic of Korea. ✉e-mail: hrkim@ncc.re.kr; cdjun@gist.ac.kr

APCs. Surprisingly, we discovered that the components of the TISs are greatly different from those identified in the T cell body, and they contain large portions of membrane components, which are essential for TCR signaling and the regulation associated with immune processes. Paradoxically, however, cells that released these essential proteins rapidly recovered surface TCRs and dramatically enforced metabolic rewiring of fatty acids synthesis (FAS), glycolysis, oxidative phosphorylation (OXPHOS), and cholesterol synthesis, leading to multiple divisions and cell survival. By contrast, a condition that internalizes TCRs augmented fatty acid catabolism via β-oxidation (FAO), which inhibited cell cycle progression. These findings highlight trogocytic molting as an intrinsic mechanism that regulates clonal T cell expansion.

## Results

### T cell adhesion on cognate APCs triggers TCR internalization and TCR release

In vitro, T cells can become activated by several mechanisms, including the use of soluble (sAb) or plate-immobilized (iAb) anti-CD3/CD28 antibodies, causing a significant downregulation of surface TCRs[14]. To corroborate surface TCR downregulation, CD3[+] T cells were stimulated with sAb, followed by secondary cross-linking, or immobilized iAb. Surface TCRβ expression was reduced in CD3[+] T cells stimulated with either sAb or iAb (Fig. 1a).

To monitor whether surface TCRs are endocytosed after T cell activation, the same T cells were pre-stained with fluorescence-labeled antigen-binding fragments (Fab), and the cells were stimulated with sAb or iAb (Fig. 1b). Interestingly, TCRβ[+] fluorescence intensity was significantly reduced in T cells stimulated with iAb, whereas it remained unchanged in T cells treated with sAb (Fig. 1b). The reduction of TCRβ[+] fluorescence intensity in iAb condition was not blocked by the endocytosis inhibitors chlorpromazine (clathrin-dependent) and PP2 (clathrin-independent), the proteasome inhibitor MG132, or the lysosome inhibitor chloroquine (Supplementary Fig. 1), suggesting that a significant number of TCRs are released during T cell adhesion on iAb, although some are internalized. We also applied flotillin 1 (flot1) siRNA to test whether flotillin-dependent pathway is involved in the molting process of TCR[+] microvilli. Knockdown of flot1 did not affect the TCR[+] shedding process and pre-stained TCRβ-fluorescent intensity changes induced by iAb or sAb (Supplementary Fig. 2).

In fact, T cells become active only after physical contact with cognate antigen-bearing APCs. This suggests that several previously conducted studies overlooked the physiological meaning of TCR release in T cell biology. Therefore, we monitored the trafficking of TCRβ during T cell activation. Similar to previous studies[1,5,14–16], TCRβ[+] fluorescent signals were dramatically internalized in T cells treated with sAb (Fig. 1c). However, a large portion of TCRβ[+] fluorescent signals was released from T cells placed on iAb as particles and spread across the T cells on the plate (Fig. 1c). Interestingly, membrane-specific green plasma membrane stain (PMS) was overlapped with TCRβ[+] particles, whereas it was internalized in T cells treated with sAb (Fig. 1c, arrowheads). The surface TCRβ release was also reproduced on the planar lipid bilayers presenting peptide-MHC (p-MHC, ovalbumin [OVA]323–339/I-A^b), which is specifically recognized by OTII TCR CD4[+] T cells, and recombinant intercellular adhesion molecule (ICAM) −1. In the absence of ICAM-1, TCRβ[+] fluorescent signals remained on the OTII CD4[+] T cell surface with minimal internalization and release. However, large amounts of TCRβ[+] fluorescent signals were spread out and detected over the plate in a density-dependent manner of ICAM-1 (Fig. 1d). To rule out suspicion of anti-TCR Fab artifacts that could be internalized or released in live cells, the similar experiments were performed expressing GFP-tagged TCR-ζ chain (ζ-GFP) in OTII CD4[+] T blasts (Supplementary Fig. 3a). As similar to the anti-TCR Fab, TCR ζ-GFP signals were significantly reduced in OTII CD4[+] T blasts on iAb but

not sAb (Supplementary Fig. 3a). Under physiological settings, DCs also acquired a significant amount of TCR ζ-GFP signals from OTII T cells, and this event was entirely dependent on the adhesion of T cells (Supplementary Fig. 3b, cyan arrowheads, and Supplementary Movies 1–3). To validate the in vivo evidence of antigen-dependent TCRβ release, wild-type C57BL/6 mice pre-immunized for 48 h with either OVA257-264 (MHC class I) or OVA323-339 (MHC class II) peptides were injected with OTII TCR T cells stained with CTV and anti-TCRβ-Fab-Alexa 488. Significant downregulation of TCRβ[+] intensity was only observed in OTII TCR T cells from OVA323-339- but not OVA257-264-immunized mice, demonstrating that membrane TCRs are released into the cognate DCs in vivo (Fig. 1e)[13]. Taken together, these results suggest that TCR release is involved in TCR downregulation at the surface of activated T cells, although some are internalized as is known, as depicted in Fig. 1f (i) and (ii).

### TCR[+] particles are shed from membrane microvilli during physical contact on immobilized activation matrix

To analyze which pathways TCRs are released from activated T cells, we performed 3-dimensional (D) confocal analysis using TCR ζ-GFP in CD4[+] T blasts or Jurkat T cells. We observed that TCR ζ-GFP predominantly localized to the thin microvilli region of T cells, with some presence in the cytoplasm (Fig. 2a and Supplementary Fig. 4a). The apical surface of microvilli consists of lipid rafts[17], which are sphingomyelin- and cholesterol-enriched membrane signaling domains and are associated with TCR signaling[17,18]. In the resting state, TCRs (or TCR ζ-GFPs) and cholera toxin B subunit (CTB), which clusters ganglioside GM1 and creates lipid rafts[19], were significantly overlapped, mostly remaining on the cell surface (Supplementary Fig. 4b and Supplementary Movie 4), whereas most were internalized under sAb condition (Fig. 2a, Supplementary Fig. 4b and Supplementary Movie 5). In contrast, both were released and spread around the T cells on iAb (Fig. 2a, Supplementary Fig. 4b and Supplementary Movie 6). Further, each microvillus contains F-actin bundles[20,21] and ERMs (ezrin, radixin, and moesin)[22]. TCRβ[+] fluorescent signals were co-localized with ezrin and F-actin at resting (NT = no treatment) state but were dissociated from ezrin and F-actin (Fig. 2b) through internalization by sAb. However, they were released through the F-actin/ezrin-enriched microvillar protrusions of activated T cells on iAb (Fig. 2b).

T cells are known to produce various EVs[7,9–11,13]. However, the adhesion-dependency of TCR release with PMS (membrane) suggests that exosomes, which are produced from multivesicular bodies, are not likely involved in surface TCR downregulation[21]. In this regard, we previously reported that T cell microvilli are highly fragile membrane structures that can be extracted when two cells split, and are transformed into nano-scale particles (i.e., TMPs) by the actions of a membrane-budding complex, such as TSG101 and Vps4a/b[13]. However, because the origins of TCR-enriched microvesicles[10] and synaptic ectosomes[11] are somehow overlapped with TMPs, we termed the TCR[+] membrane particles released from the activated T cells on adhesion substrates as TISs.

Scanning electron microscopy (SEM) analysis showed that, after initial extension to the iAb surface (Fig. 3a, iAb, -1 min=initial and early), microvilli remained at the most distal edge of distal-supramolecular cluster (d-SMAC) during maturation of IS (∼5 to 10 min=mature), and then released from T cell body at terminal stage in the form of large rod-shaped TISs (∼60 min=terminal) (Fig. 3a). A schematic model of particle generation is depicted in Fig. 3a. Similar results were also observed in mouse OTII CD4[+] T cells placed on lipid bilayers presenting OVA323–339/I-A^b/ICAM-1 (Fig. 3b, OTII CD4[+] T cells in kinapse). Un-fragmented large rod-shaped particles were further fragmented (Fig. 3b, arrowheads). Using the microvilli-specific protein Vstm5 tagged with GFP (V5G, Supplementary Fig. 5a–c), we corroborated that long membrane protrusions are fragmented at the rear edge of migrating cells (Fig. 3b, OTII CD4[+] T cells in kinapse; Supplementary

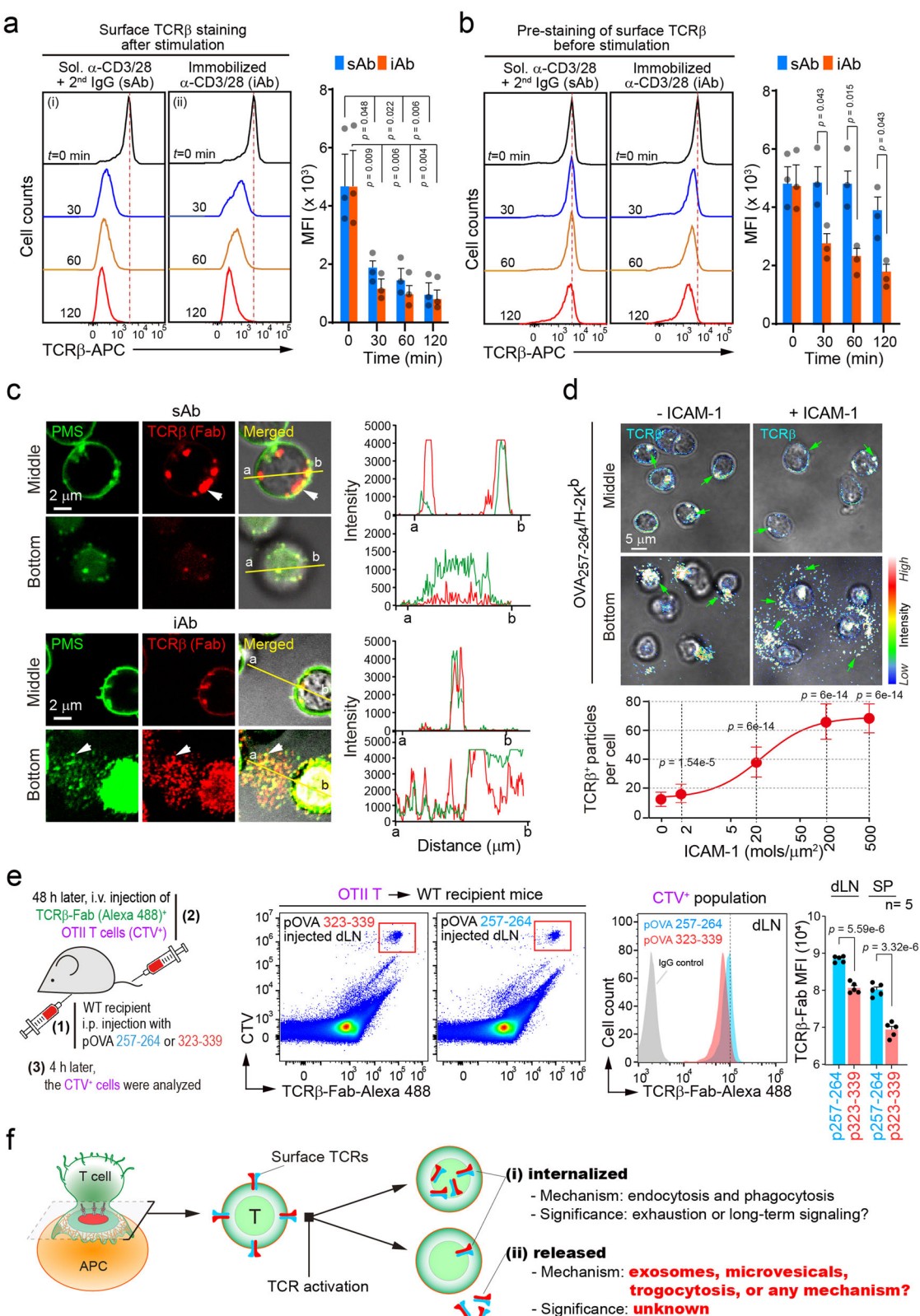

Movie 7). A schematic model describing particle generation from protrusive rear edges during T cell kinapse is shown in Fig. 3c. Consistently, TCR downregulation correlated with the loss of microvilli, as determined by SEM (Fig. 3d) and flow cytometry (Fig. 3e). Remarkably, only TCRβ and CD3 complexes were reduced by sAb in *OTII* CD4 T cells, whereas several microvilli-specific molecules, including CD62L, GM1, and CD4 were also reduced by iAb (Fig. 3e)[22]. Both CD25 and CD4

were not significantly reduced compared to TCR, suggesting that they are not fully co-localized with TCRs in the microvilli[22]. Consistent with this line, the CD4-fluorescence intensity in the cell membrane was not significantly reduced compared to TCRβ intensity (Supplementary Fig. 6). A decrease of microvilli-specific molecules was also observed in *OTII* CD4⁺ T cells on OVA₃₂₃₋₃₃₉/I-Aᵇ/ICAM-1, whereas they were minimally reduced in the absence of ICAM-1 (Fig. 3e).

**Fig. 1 | T cell activation triggers TCR internalization and TCR release.** Naïve CD3[+] T cells were stained with anti-TCRβ-APC after (**a**) or before (**b**) stimulation with soluble (sAb) or plate-immobilized (iAb) anti-CD3/CD28 antibodies for 3 h. The mean fluorescence intensity (MFI) was measured by flow cytometry. Data represent the mean ± SEM of three independent experiments. **c** CD3[+] T cells stained with the CellMask™ Green Plasma Membrane Stain (PMS) and anti-TCRβ-Alexa594 for 1 h at 4 °C were stimulated as in (**b**). White arrows indicate internalized or released TCRβ clusters. These results were independently repeated three times. **d** CD8[+] T cells from an OTI mouse were stained as in (**b**) and placed on a lipid bilayer presenting pOVA$_{257-264}$/H-2K$^b$/ICAM-1 (0–500 molecules/μm$^2$) for 3 h. Green arrows indicate

internalized or released TCRβ[+] clusters or particles. **e** Schematic diagram of in vivo experimental setting for antigen-dependent TCRβ release. Wild-type mice were i.p. administered with either 100 μg of pOVA$_{257-264}$ or $_{323-339}$. After 48 h, TCRβ-Alexa488 and CTV stained naïve OTII CD4[+] T cells were i.v. injected. TCRβ intensity of CTV[+] T cells from draining lymph node and spleen was determined by flow cytometry after 4 h. **f** Schematic model of TCR internalization and release. The model hypothesizes that not all TCRs are internalized (i), but some are released (ii) during T–APC interaction. Statistics were performed using unpaired two-tailed *t* test (**e**) or one-way ANOVA with post hoc Tukey's multiple comparisons test (**a**, **b**, **d**). Source data are provided as a Source Data file.

Lipid compositions enriched in TISs and total cell extracts were determined. Based on log fold changes, we identified substantial enrichment of specific lipid species in TMPs, including sphingomyelin (SM) and phosphatidylserine (PS), which were known to be enriched in lipid rafts and exosomes (Kolmogorov-Smirnov tests, *p* values < 1 × 10$^{-5}$, Supplementary Figs. 7a and 7b). Notably, we also found that more enrichment of saturated phospholipids compared to unsaturated counterparts, including phosphatidylglycerol (PG), phosphatidylinositol (PI), and lysophosphatidic acid (LPA), which would increase lipid packing density in TISs (Wilcoxon tests, *p* values < 0.05).

Because TIS generation requires both trogocytosis (activation-induced T cell adhesion) and enzymatic breakage (vesiculation or shedding), we called this event "trogocytic-molting" or "trogocytic-shedding." Collectively, these results led us to investigate the physiological significance of TCR loss after the trogocytic-molting of microvilli compared with TCR internalization at the initial stage of naive T cell activation.

## Molting is associated with TCR expression and metabolic upregulation

Another long-known concept of T cell stimulation methods is that plate-bound, immobilized anti-CD3/28 (iAb) produces a stronger signal than soluble anti-CD3/28 (sAb). To prove this concept in terms of TCR signaling intensity, we examined the phosphorylation of downstream signals after the stimulation of TCR under soluble or immobilized conditions. Intriguingly enough, we observed that the stimulus intensities were nearly similar in terms of phosphorylation levels of TCR downstream molecules (Fig. 4a and Supplementary Fig. 8). In addition, there was no difference in the levels of phosphorylated protein kinase θ (PKCθ), extracellular signal-regulated kinase, and p38 kinase until 3 h of stimulation (Supplementary Fig. 9a). In contrast, the only significant difference between sAb and iAb we found was the number of generating TCRβ[+] particles (Fig. 4b). The number of TCRβ[+] TISs released were almost reached a maximum within 5–10 min in the iAb condition (Fig. 4b). On the contrary, these particles were not seen in the sAb condition, probably due to internalization (Fig. 4b). These results suggest that the "trogocytic-molting" process is a very rapid process and can be completed within ~5–10 min.

Since TCR[+] release via TISs or TCR internalization occurs only within 5–10 min in activated T cells, we initially chose 30 min for TCR stimulation in both conditions. Naive T cells were stimulated for 30 min (0.5 h) with either iAb or sAb, then washed (0 h), and further cultured for the indicated time periods (0–72 h) with fresh media. We first investigated the shift in the size and granularity of T cells and found a significant difference in their morphology between iAb and sAb conditions (Fig. 4c). To explore whether TCR expression levels correlated with the number of microvilli and were proportional to the cell size and granularity, T blasts at 72 h of culture were sorted into two groups (i and ii) and analyzed under scanning EM. Group (i) T blasts were similar in the number and size of microvilli as those from sAb (Fig. 4c). However, group (ii) T blasts were larger, contained a greater number of microvilli (Fig. 4c), and exhibited higher expression of surface T cell proteins than those from group (i) (Supplementary Fig. 9b). T blasts could release more TISs, presumably because of more

microvilli (Supplementary Fig. 10). Interestingly, in contrast to sAb treatment, T cells treated with iAb displayed a rapid recovery of surface TCRβ (Fig. 4d). Along together with cell size increase and TCR recovery, a dramatic increase in cell division (Fig. 4e) and the proliferation indicator Ki-67 (Fig. 4f) and reduced apoptotic cell death (Fig. 4g) were observed in T cells treated with iAb. In contrast, increased apoptotic cell death was seen in T cells treated with sAb. No qualitative difference was observed in TCR recovery and proliferation in CD4[+] and CD8[+] T cells (Supplementary Fig. 11a, b).

T cell activation results in the metabolic remodeling of naive T cells to a program of aerobic glycolysis, which can generate metabolic intermediates essential for cell growth and proliferation[23]. We compared the metabolic changes of naive T cells between the two conditions. To mimic the iAb condition, we utilized CD3/CD28-coated dynabeads (dAb). A real-time analysis revealed that the oxygen consumption rate (OCR) of T cell stimulation was not significant in both dAb and sAb conditions. However, the dAb-activated T cells showed a dramatic increase in the extracellular acidification rate (ECAR), which is an indicator of aerobic glycolysis, within 30 min, suggesting that the dAb-activated T cells produced energy through a process of glycolysis at a higher rate than the cells in the sAb condition (Fig. 4h). The time interval at which TCRβ[+] particles are maximally released appeared to be correlated with the time interval at which T cell metabolism increases (Fig. 4h). To determine whether the pattern of changes in ECAR and OCR persists over time after stimulation, ECAR and OCR levels were measured after 24–72 h of culture. We observed higher ECAR and OCR levels in the activated T cells by iAb than by sAb (Fig. 4i), indicating distinguished metabolic reprogramming after TCR loss at the T cell surface. In addition, increased synthesis of *TCRα* gene was seen after 1 h of iAb stimulation (Fig. 4j), suggesting that rapid recovery of surface TCR is due to the de novo synthesis of *TCR* genes.

## Molting of membrane microvilli is critical for T cell clonal expansion

In pursuit of direct evidence that trogocytic microvilli shedding is truly critical for T cell clonal expansion, we developed several strategies. First, to exclude a potential artifact induced by the plate-immobilized antibody, we chose ICAM-1 as an adhesion matrix to induce T cell adhesion. Second, because LFA-1, a ligand of ICAM-1, also has a costimulatory function, we tested anti-CD62L (L-selectin) as a microvilli-specific adhesion matrix. Third, we investigated whether T cell activation by sAb also can be increased under even nonspecific adhesion condition. To this end, the plate was coated with normal (1× = 5 × 10$^{-4}$%) or a higher concentration (10× = 5 × 10$^{-3}$%) of poly-L-lysine (c-PLL) to immobilize sAb-activated T cells on the culture plate. Forth, to rule out any artifact of non-specific immobilization of sAb on the culture plate, naive CD3 T cells were stimulated only for 5 min, as this time period was sufficient to fully release TISs, and then washed and placed on the above mentioned adhesion matrices. Unlike sAb alone, surprisingly, the plating of pre-stimulated naive T cells with sAb (5 min) on the surface of any adhesion matrix could significantly release PMS[+] TISs (Fig. 5a). Moreover, the increased release of TISs correlated with the significant induction of T cell proliferation (Fig. 5b). Interestingly, similar to the results under iAb conditions, metabolic reprogramming

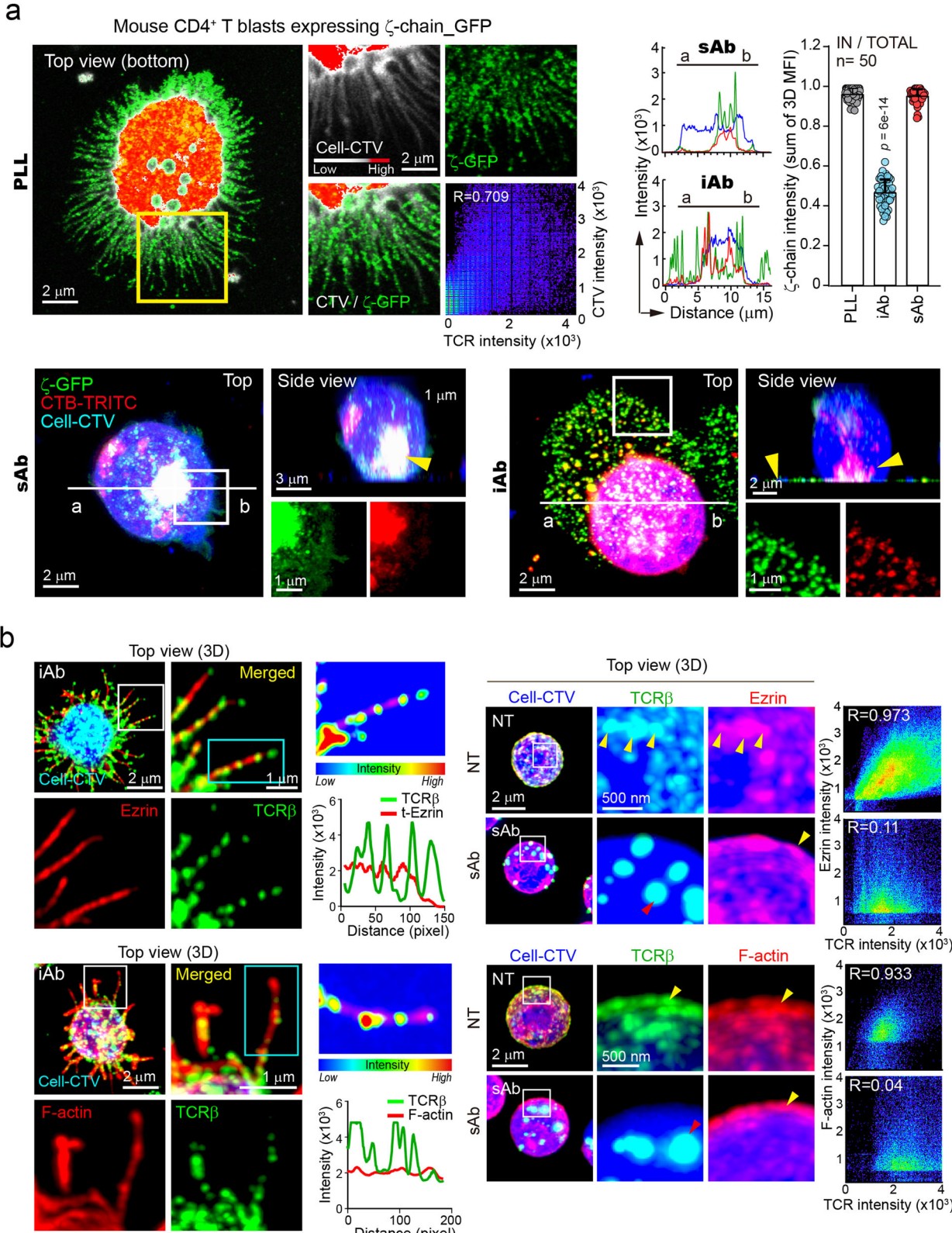

was dramatically increased (Fig. 5c), followed by rapid induction of surface TCR (Fig. 5d). As shown in Fig. 5e, T cell proliferation was clearly correlated with the amount of released membrane particles (PMS⁺).

In another experiment, DCs were lightly fixed with 0.4% paraformaldehyde to provide only adhesion sites for T cells. This condition also mimicked the matrix-coated surface (Fig. 5f, g). Further, we

examined other T cell stimuli such as concanavalin A (ConA) and phorbol 12-myristate 13-acetate (PMA)/ionomycin instead of antibodies. ConA is known to irreversibly bind to glycoproteins on the cell surface and commit T cells to proliferation[24]. Interestingly, ConA alone had little effect on TCR internalization, whereas it significantly induced the release of TCRβ⁺ TISs on the coated matrices (Fig. 5h). Consistently, T cell proliferation was enhanced (Fig. 5i). Similar results were also

**Fig. 2 | TCRs are released through the F-actin/ezrin-enriched microvillar protrusions of activated T cells on iAb. a** CD4[+] T blasts expressing TCRζ_GFP were stained with either CTV or CTB (cholera toxin B) and stimulated with iAb (10 µg/mL) or sAb (10 µg/mL) for 3 h. Localization of TCRζ_GFP in resting (PLL, top) or activated conditions (TCRζ_GFP or CTB, bottom) was observed. Yellow arrows represent internalized or released TCRζ_GFP at each condition. TCR ζ intensity (sum of 3D MFI) was presented as inside intensity divided by total intensity. For each single experiment, 50 cells were randomly selected and analyzed. Results are representative of three independent experiments ± SD. Statistics was performed using one-way ANOVA with post hoc Tukey's multiple comparisons test vs. PLL. Source

data are provided as a Source Data file. **b** CTV and anti-TCRβ-Alexa488-pre-stained naive CD3[+] T cells were stimulated as in (**a**), fixed, permeabilized, and stained with either anti-ezrin antibody or phalloidin-TRITC. TCRβ localization across the microvilli in boxed area (cyan) was represented with pseudo-color coding according to fluorescence intensity. Yellow arrows represented colocalization of TCRβ with either ezrin or F-actin in resting condition. Red and yellow arrow represented dissociation of TCRβ from ezrin and F-actin in sAb condition. All images were acquired by AiryScan confocal microscopy and reconstructed for the 3D by using Zeiss ZEN software. These results were independently repeated three times on more than 10 randomly selected cells each time.

observed with PMA/ionomycin (Supplementary Fig. 12). Taken together, the current results strikingly suggest that, at least in vitro, TCR signaling only for ~5 min is sufficient to activate T cells, but T cell proliferation necessarily requires a molting process regardless of any adhesion matrix.

We also examined the relationship between molting process and T cell proliferation in more physiological conditions. Although adhesion matrix such as ICAM-1 and VCAM-1, but not fibronectin (FN), slightly increased TCRβ[+] particles, OVA$_{323-339}$/I-A[b] (pMHC) in combination with ICAM-1 or VCAM-1 significantly increased the release of TCRβ[+] particles (Fig. 6a). As expected, surface TCR loss is completely correlated with the TCRβ[+] TIS release. Further, the cells exhibiting surface TCR loss after TIS release entered the cell cycle for proliferation (Fig. 6b).

If the reason for the increase in T cell metabolism depends on microvilli shedding rather than adhesion itself, the reagents that induce microvilli release or disruption may enhance the T cell metabolic programming for proliferation under sAb condition. We used various inhibitors that interfere with the actions of the cytoskeleton or endocytosis. Among the various inhibitors, latrunculin A (LatA) and cytochalasin D (CytD), both actin-depolymerizing reagents, significantly reduced the number of microvilli (Fig. 7a). Interestingly, this result is consistent with the previous reports demonstrating that actin-disrupting agents can rapidly release the coated vesicles from the plasma membrane[25,26].

In contrast, CK636 (an Arp2/3 complex inhibitor), jasplakinolide (JPK; an actin-stabilizing agent), and other inhibitors had little effect (Fig. 7a). To this end, T cells were stimulated with 5 min, then washed, and further cultured with various inhibitors including LatA and CytD. To our surprise, treatment of LatA or CytD in T cells activated with sAb (5 min) dramatically induced PMS[+] TIS release, whereas JPK and CK636 had little effect (Fig. 7b). Consistently, the molted T cells showed dramatic proliferation (Fig. 7b) with the enforced metabolic reprogramming (Fig. 7c). These results suggest that surface microvilli release is linked with metabolic reprogramming for T cell clonal expansion. However, the lack of effect of various endocytosis process blockers suggests that endocytosis of TCR does not correspond to T cell proliferation.

To further obtain evidence that microvilli separation via trogocytic-molting is a direct mechanism of T cell proliferation, we utilized *LFA-1*[-/-] (LFA-1 KO) T cells. Previously, we reported that T cells from *LFA-1*[-/-] mice do not adhere to mICAM-1-coated plates after α-CD3/28 stimulation[13]. Indeed, LFA-1 KO T cells released fewer PMS[+] TISs than wild-type (WT) T cells when placed on ICAM-1 after sAb stimulation (5 min) (Fig. 7d). However, these T cells significantly induced PMS[+] TISs under the conditions of anti-CD62L, PLL (10×), and LatA (Fig. 7d). Again, LFA-1 KO T cells that underwent TIS release via trogocytic-molting strongly entered the cell cycle for proliferation (Fig. 7d).

Long, rod-shaped microvilli particles are further vesiculated by the actions of a budding complex, such as Vps4a/b[13]. In fact, TIS contains higher amounts of Vps4a/b (Fig. 8a). We investigated whether knockdown of Vps4a/b may retard the TCR recovery and T cell proliferation. Si-RNA targeting Vps4a/b significantly reduced the

production of TISs on OVA/H-2K[b]/ICAM-1 (Fig. 8a). After stimulation on the lipid bilayer (OVA/H-2K[b]/ICAM-1), although TCRβ loss on the surface of OTI T cells was less than that of the scrambled si-RNA-transfected T cells, TCRβ recovery was significantly delayed (Fig. 8b). Similarly, attenuated metabolic reprogramming, followed by a reduced Ki-67[+] cell population, was observed in T cells (Fig. 8c). Interestingly, CD25 and CD69 expression was not altered by si-Vps4a/b (Fig. 8d), indicating that the reduction of Vps4a/b has no effect on initial T cell activation.

If actin disruption induces microvilli shedding, then actin stabilization may inhibit microvilli fragmentation, thereby blocking TCR loss. To this end, we tested whether several actin drugs used in this study could affect T cell activation. CK636 and JPK had little effect on T cell spreading and subsequent activation on the OVA/H-2K[b]/ICAM-1 (Supplementary Fig. 13). Interestingly, the actin stabilizer JPK inhibited the release of TCRβ[+] TISs induced by OVA/H-2K[b] ICAM-1 (Fig. 8e) and delayed cell proliferation (Fig. 8f), while it did not alter the downstream signals of TCR until 3 h of stimulation (Supplementary Fig. 9a). Notably, LatA which enhanced T cell proliferation in sAb-activated T cells (Fig. 7b) significantly reduced T cell spreading on the OVA/H-2K[b]/ICAM-1 (Supplementary Fig. 13) and subsequent T cell activation (Fig. 8g). In addition, LatA inhibited TCRβ[+] TIS release induced by OVA/H-2K[b]/ICAM-1, suggesting that intact F-actin structure, i.e., microvilli composed of F-actin, is critical for T cell antigen recognition and subsequent T cell activation. However, after proper TCR stimulation, a molting process is essential for T cell proliferation.

### TISs exclusively contain external membrane components involved in immune and metabolic processes

To elucidate a potential mechanism for how trogocytic molting of microvilli enhances T cell proliferation, purified TISs obtained from activated naive CD3[+] T cells were subjected to liquid chromatography-tandem mass spectrometry (LC-MS/MS). Proteins that were commonly detected from three independent experiments on the total cell lysates (TCLs) and TISs were selected (Fig. 9a). This analysis revealed 3537 (80.7%) proteins in the TCLs and 3216 (74.4%) in TISs, of which 2375 (54.2%) were common in both samples. The high degree of overlapping proteins observed between TCLs and TISs indicated that TISs contain most of the TCL proteins (Fig. 9a). However, a cellular component analysis based on Gene Ontology annotation revealed that TIS-only exclusively contains external membrane components while TCL-only includes a variety of internal cellular components (Fig. 9b). In addition, analysis and visualization of biological processes with Cytoscape demonstrated that proteins included in TIS-only are related to the regulation of immune processes and vesicle transport while those in TCL-only play a variety of roles in cellular pathways (Fig. 9c).

We further analyzed the cellular components and biological processes of the proteins commonly found in both groups (2,375 proteins). To this end, particular repertoires of proteins were selectively sorted into TCL or TIS based on twofold difference by relative quantitation provided by the exponentially modified protein abundance index (emPAI) (Fig. 9d). Similar to the results of TIS-only, cellular component analysis revealed that proteins enriched in TISs are derived from membrane components (Fig. 9e). Interestingly, analysis of the

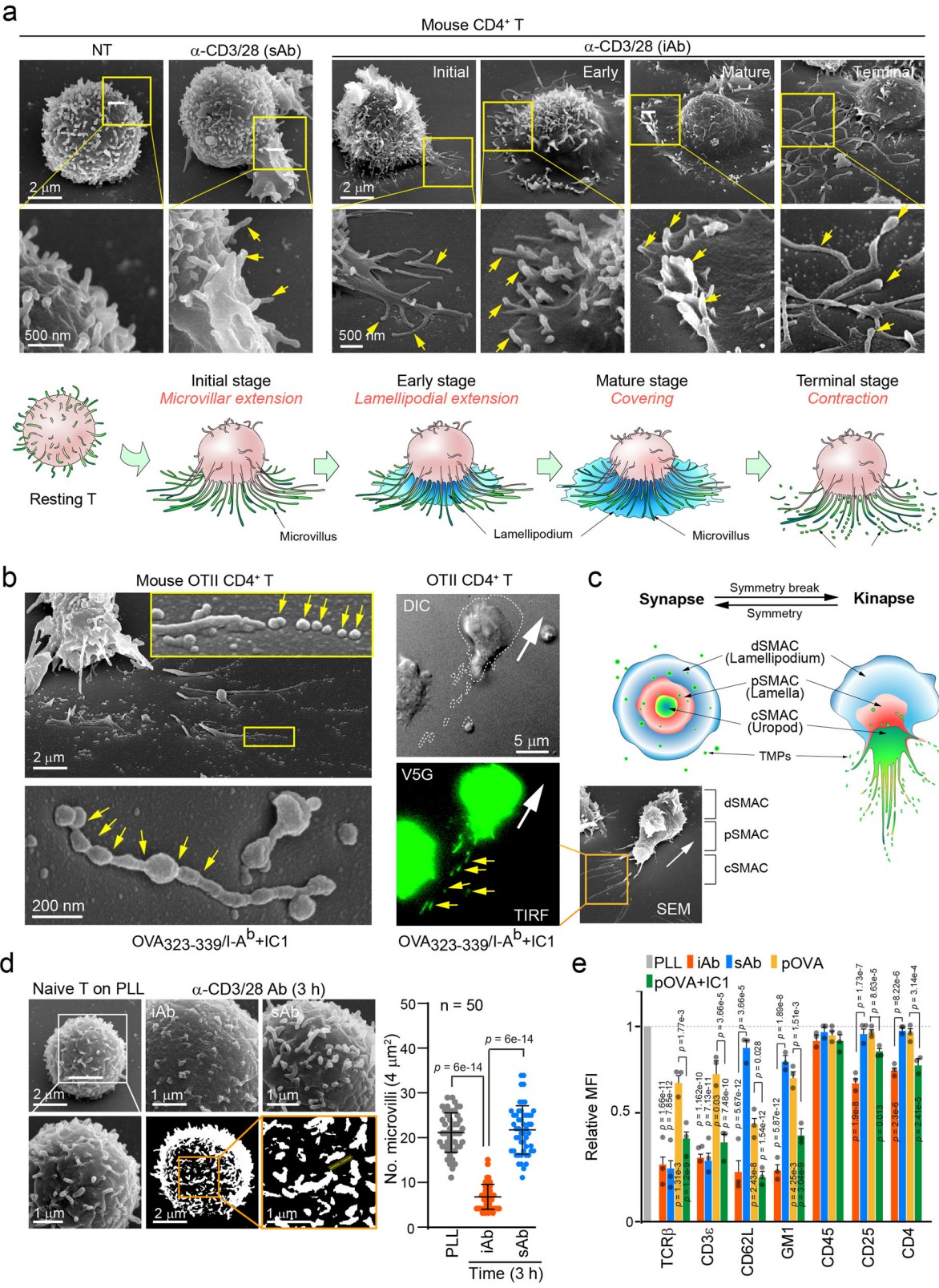

biological process revealed that, in addition to protein involvement in the regulation of immune function, TISs contained proteins that are related to energy metabolism, such as ATP hydrolysis, oxidation-reduction process, and lipid metabolic process (Fig. 9f). From the analysis of proteins in TISs, we further hypothesized that the release of external membrane components such as TCR complex and many T cell proteins, cholesterols and glycolipids, and some metabolic proteins

via shedding of T cell microvilli may evoke a significant alternation in the process of T cell activation, and hence T cell fate.

## T cell molting favors FAS, whereas TCR internalization favors FAO

To understand whether the release of membrane components, including TCRs, via the molting of microvilli regulates gene expression

**Fig. 3 | T cells shed surface microvilli when they bind to the adhesion matrix during activation by TCR signaling. a** Scanning Electron Microscopy (SEM) images of naive CD3[+] T cells stimulated with sAb for 3 h or iAb at various time points. Initial (<1 min), early (1 min), mature (1–5 min), and terminal stages (>60 min). A schematic model of TISs generation during T cell activation is presented. These results were independently repeated three times from more than 20 randomly selected cells (NT, α-CD3/28 (iAb), respectively), and similar results were consistently obtained in all replicates. **b** SEM evidence of TISs release on the lipid bilayer presenting pOVA$_{323–339}$/I-A$^b$/ICAM-1 (left). Yellow arrows indicate spherical TISs before and after fragmentation from T cell microvilli. Release of V5G[+] microvilli particles from OTII CD4[+] T cells during migration on pOVA$_{323–339}$/I-A$^b$/ICAM-1 (right). These results were independently repeated three times from more than 20 randomly selected OTII CD4[+] cells, and similar results were consistently obtained in all replicates **c** A schematic model describing TISs generation during T cell kinapse. **d** Naive CD3[+] T cells were stimulated with iAb or sAb and cell surface was observed by SEM. The number of microvilli per 4 μm$^2$ was quantitated by using the ImageJ software and FiloQuant plugin. For each single experiment, 50 cells were randomly selected and analyzed. Data represent the mean ± SEM of three independent experiments. Source data are provided as a Source Data file. **e** Flow cytometric analysis of surface proteins potentially enriched or excluded in microvilli after the stimulation of naive CD3[+] T cells as in (**d**). Data represent the mean ± SEM of three independent experiments. The significance was indicated on the bar graph. Statistics was performed using one-way ANOVA with post hoc Tukey's multiple comparisons test (**d**, **e**). Source data are provided as a Source Data file.

involved in certain pathways, we generated the transcriptome of naive T cells stimulated with sAb, iAb, LatA, and sAb plus LatA (sAb+LatA) through expression profiling of the microarray (Affymetrix Mouse Gene ST 2.0). Based on hierarchical clustering, we observed the coordinated expression changes by time (baseline, 3 h, and 24 h) and group (naive T, LatA, sAb, sAb+LatA, and iAb-activated T cells) (Fig. 10a). Based on principal component analysis (PCA), we observed the trajectory of expression changes from naive T to LatA, sAb, sAb+LatA, and iAb T cells (red arrow) (Fig. 10b). Focusing on naive T cells compared with sAb- and iAb-treated T cells (Fig. 10c), we performed pathway enrichment tests (R gProfileR package) between iAb and sAb (both 3 and 24 h) (Fig. 10d). Interestingly, we found that the cellular metabolic process was significantly changed at 3 h, whereas the cell cycle process was significantly changed at 24 h (enrichment test $p$ values < 10$^{-8}$). Therefore, we speculated that iAb-activated T cells could undergo dramatic metabolic reprogramming at 3 h, and later its reprogramming affected the cellular proliferation at 24 h.

The dramatic changes in the cellular metabolic process (Fig. 10d) led us to perform in-depth metabolic pathway enrichment tests using high-quality metabolic maps[27,28]. Based on normalized enrichment scores (NES) of expression changes (R fgsea package), we detected distinct clusters of metabolic pathways enriched in differently by conditions. First, we observed the pathways of maximal expressions in iAb-activated T cells, including FAS, OXPHOS, glycerolipid, cholesterol, and nucleotide metabolism, which could promote the recovery of membrane loss and lipid rafts (Fig. 10e). GPI-anchor synthesis, which is also involved in the formation of lipid rafts, was highly expressed in both sAb- and iAb-activated T cells compared with naive T cells (Fig. 10e). Cholesterols and fatty acids are a vital component of cell membranes, can supply substrates for cell signaling, and provide a high-yielding energy source[23,29]. Significant upregulation of lipid metabolism pathways (Fig. 10e), together with the increased glycolysis and OXPHOS pathways (Figs. 4h and 4i), in iAb- *vs.* sAb-activated T cells strongly suggest that membrane microvilli molting is evidently linked with metabolic processes for T cell proliferation. Second, we detected the following two distinct pathway clusters specifically changing in sAb-activated T cells: (1) a decreased cluster that includes B12 metabolism, a cofactor of DNA synthesis, and (2) an increased cluster that includes FAO, a metabolic feature of memory T cells. Vitamin B12 deficiency reduces proliferation in some cell types[23,29]. Therefore, increased FAO and reduced B12 pathways in sAb-activated T cells suggest that the TCR internalization event is linked to slowing or halting of cell cycle progression. The representative genes clustered in the FAS or FAO pathways are shown in Fig. 10e. These findings prompted us to investigate the rate of fatty acid oxidation (FAO) by using either palmitate-BSA or by measuring the levels of three key enzymes involved in β-oxidation: medium-chain acyl-CoA dehydrogenase (ACADM), very long chain acyl-CoA dehydrogenase (ACADVL), and the alpha subunit of the trifunctional multi-enzyme complex (HADHA). sAb-activated T cells demonstrated a marked rise in OCR when exposed to palmitate and displayed higher levels of the crucial FAO enzymes compared to iAb-activated T cells. These results

indicate that iAb-activated T cells rely less on lipids as an energy source, whereas sAb-activated T cells exhibit a stronger preference for FAO relative to iAb-activated T cells (Fig. 10f and g). Interestingly, increased expression of metabolic pathways was also observed in sAb +LatA-activated T cells compared with sAb treatment (Supplementary Figs. 14a and b).

We also generated the transcriptome of naive T cells and ConA (low-dose)-, PLL (10×)-, and ConA-PLL-treated T cells and found that only ConA+PLL, which dramatically release TCR[+] TISs, induced the upregulation of cell proliferation (enrichment test p values < 10$^{-8}$) (Supplementary Figs. 15a and 15b) and cellular biomass synthesis (e.g., cholesterol, nucleotide, amino acids, and fatty acids) (Supplementary Figs. 15c and 15d; fGSEA p values < 0.05). Interestingly, genes associated with FAO were not significantly regulated with ConA treatment, presumably because ConA did not cause internalization of the TCR complex (Fig. 5h).

Collectively, the present study demonstrates that the physical interaction of T cells with their cognate APCs is physiologically significant in two manners. First, as shown in previous studies, T cell activation induces TCR internalization, which can induce "biomass conversion into energy," thereby slowing or halting T cell proliferation. Second, at the initial stage of naive T cell activation, however, T cells require massive expansion to procure effector functions, and this is possibly obtained by the molting of TCR-enriched microvilli, which can induce "biomass generation" (FAs, cholesterols, glycerolipids, etc.). A schematic model based on global metabolic gene profiling and the metabolic functional changes caused by molting or TCR internalization are depicted in Fig. 10h.

## Discussion

TCR downmodulation after engagement with p-MHC is a hallmark of T cell activation and is believed to occur through increased TCR internalization or rapid degradation[14,30]. However, in the present study, we showed that molting of T cell membrane protrusions including microvilli in the form of TISs is a crucial mechanism corresponding with the loss of TCRs on the surface of activated T cells. T cell microvilli have at least two major functions. First, they are important outer-layer organelles to sense antigen peptides on APCs[12,13,21,31]. Second, during the physical interaction with APCs, T cell microvilli act as carriers to release a large number of proteins and lipids required for T cell life. Paradoxically, these T cells enter multiple division cycles and grow to meet the demands of clonal expansion by the de novo synthesis of gene clusters involved in lipid metabolic pathways, including FAS, cholesterols, glycerolipids, and lipid rafts. To our knowledge, these findings are new evidence that, apart from external factors such as cytokines released from APCs, T cells have an intrinsic program for proliferation and clonal expansion by which the T cells gain increased sensitivity to low concentrations of antigen.

Unlike the many studies focusing on TCR internalization, none has investigated whether the release of TCRs from T cells encountering their cognate APCs can affect surface TCR downregulation. Therefore, no one has examined what percentages of TCRs are internalized or

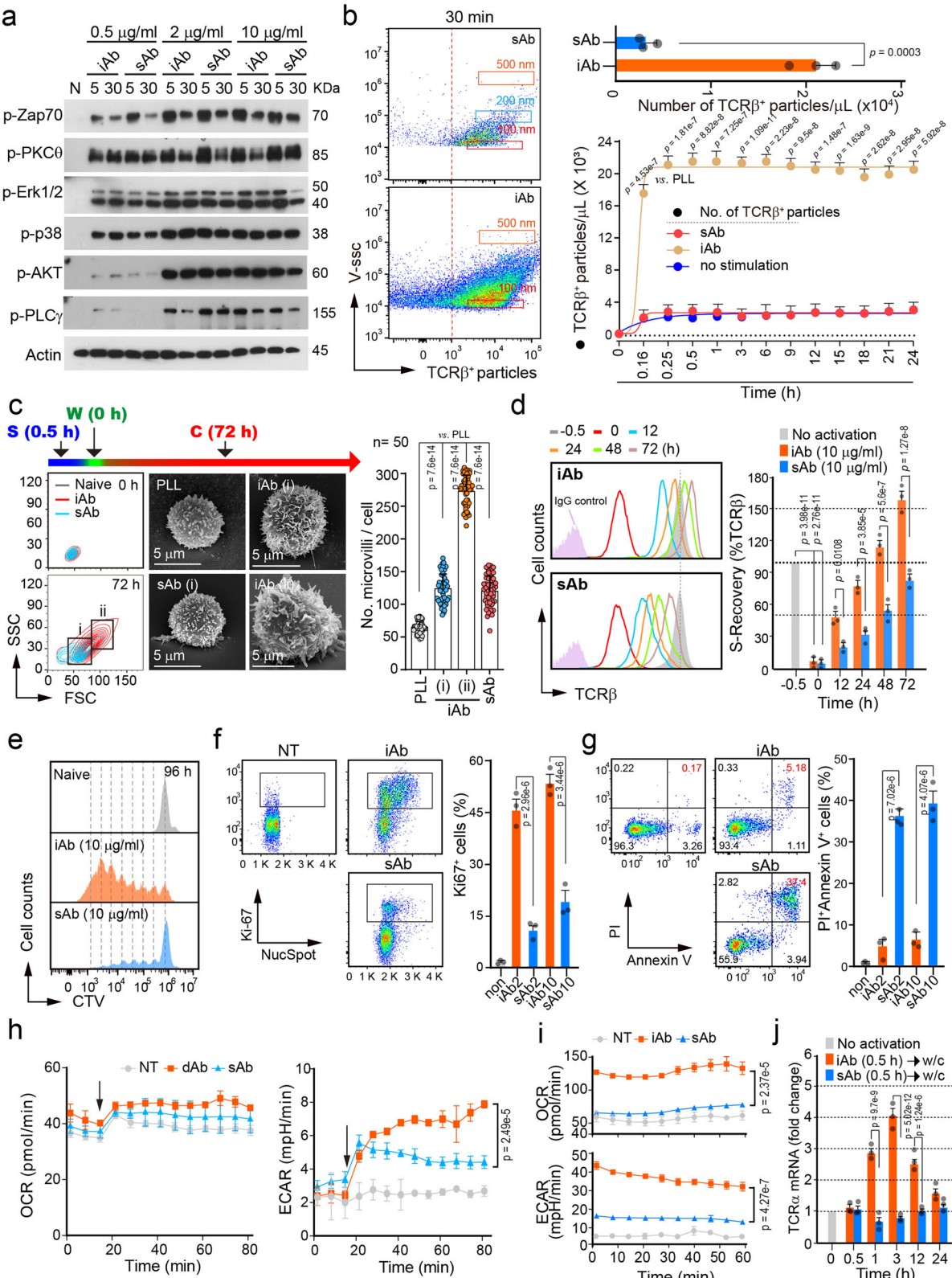

released during T cell contact with an APC. Interestingly, most TCR-triggering studies that have used surface-coated anti-CD3 antibody or peptide-pulsed APCs did not account for microvesicles or microvilli-derived particles released from activated T cells. This presupposes that all TCRs are internalized during T cell activation; thus, the results produced by this preconception are likely to be interpreted from only one perspective. The present study reveals that TCR traveling is

distinguished based on whether the stimulating conditions occur in a soluble or matrix-coated state. From this point of view, the coated surface is more similar to the physiological state. Indeed, we[13] and others[32] have clearly demonstrated that a considerable number of TCRs are transferred to the APCs during IS formation and deformation.

What are the mechanisms of TCR release from T cells encountering their cognate APCs? Unlike the generation of exosomes or

**Fig. 4 | Trogocytic-molting of T cell microvilli enforces metabolic reprogramming for T cell proliferation and survival. a** TCR signaling intensity following iAb or sAb treatment. Results are representative of three independent experiments. Densitometric analysis is presented in Supplementary Fig. 8. The samples shown are from the same experiment, with gels/blots processed in parallel. **b** Naive CD3$^+$ T cells were pre-stained with anti-TCRβ-Alexa488 and stimulated with iAb or sAb for the indicated time periods. Left, representative results at 30 min. Right, number of released TCRβ$^+$ particles quantified by CytoFLEX at the indicated time point. **c**, **d** Naive CD3$^+$ T cells were stimulated for 30 min with iAb or sAb, washed, and cultured for 72 h with fresh medium. T cells in the gates (i and ii = iAb; i = sAb) were sorted according to cell size (FSC) and granularity (SSC) and observed under SEM. The number of microvilli per cell was quantified using ImageJ (**c**). For each single experiment, 50 cells were randomly selected and analyzed. Data represent three independent experiments ± SD. Surface TCRβ recovery was determined by staining with TCRβ-Alexa 488 at the indicated time points (**d**). **e**–**g** CTV-stained naive CD3$^+$ T cells were stimulated as in (**c**), and cell division at 96 h (**e**), proliferation (Ki-67 staining) and NucSpot staining at 48 h (**f**), and apoptosis (Annexin V/PI staining) at 48 h (**g**) were determined. non, untreated; iAb2 and sAb2, 2 μg/mL; iAb10 and sAb10, 10 μg/mL. **h** Oxygen consumption rate (OCR) and extracellular acidification rate (ECAR) trace of naive CD3$^+$ T cells after stimulation with sAb or dynabead-immobilized antibody (dAb). Black arrowheads indicate stimulation point. (**i**) Naive CD3$^+$ T cells were stimulated as in (**c**) and OCR and ECAR trace were measured 24 h post stimulation. **j** TCRα mRNA expression at the indicated time points after stimulation. w/c, wash and culture. Data represent the mean of three experiments ± SD (**b**, **c**, **h**, **i**) or SEM (**d**–**g**, **j**). Statistics were performed using unpaired two-tailed t test (**a**, **c**) or one-way ANOVA with post hoc Tukey's multiple comparisons test (**b**, **d**, **f**–**j**). Source data are provided as a Source Data file.

microvesicles, microvilli shedding depends on trogocytosis[7,33,34], which was thought to transfer membrane patches containing intact molecules via enzyme-independent cleavage[34]. However, we recently identified that fragmentation (or vesiculation) of microvilli from the T cell body requires enzyme complexes such as TSG101 and Vps4a/b, which mediate the budding of microvesicles at the cell surface[35]. In an agreement with this line, knockdown of Vps4a/b significantly reduces TIS release via microvilli shedding[13]. To our knowledge, our work showed the first evidence that T cells release microvilli particles through the combined action of two independent mechanisms: trogocytosis and enzymatic breakage. Therefore, the "trogocytic-molting" of surface membrane components may be a common mechanism occurring during immune cell-to-cell interaction.

Interestingly, it is very commonly accepted that plate-bound, immobilized anti-CD3/28 is believed to work better as it mimics cell-cell contact and may also guarantee the chance for the cells to interact with immobilized anti-CD3/28 antibodies on the surface of the plate[36]. However, our current study evidently demonstrated that two conditions, i.e., soluble and immobilized, are not different in terms of the intensity of TCR signaling. Indeed, we found that ~5 to 10 min TCR activation is sufficient to enforce T cells to undergo metabolic changes and subsequent proliferation for clonal expansion. The only difference we found was that T cells release a great number of essential molecules including TCRs via the release of TISs (or TMPs) upon binding to the immobilized matrix, whereas they are internalized in sAb condition. Therefore, we argue that generally accepted notion that immobilized antibodies have stronger and longer-lasting signals than soluble antibodies is a misguided paradigm.

At least two main evidence of current study strongly demonstrated that TCRs are released via the fragmentation of membrane outer layers, i.e., microvilli. First, previous reports demonstrated that lipid rafts, which are membrane signaling domains associated with TCR signaling[17,18], are located at the apical surfaces of microvilli[17]. Interestingly, the apparent presence of CTB, a marker for GM1 in lipid rafts, with plate-bound TCR$^+$ particles released from activated T cells on iAb strongly demonstrates that TCR-enriched TISs are formed mainly from the shedding of surface microvilli. Unlike the sAb condition, a reduced number of microvilli and downregulation of microvilli-specific proteins in T cells activated via the iAb condition also provide a direct evidence that the particle origins are microvilli covering the outer layer of the cell. Second, Dustin's group reported that T cells release TCR-enriched microvesicles or synaptic ectosomes through IS while interacting with APCs[10,11]. They demonstrated that membrane particles are primarily released from the central region of the supramolecular activation cluster (c-SMAC). If TCR$^+$ particles are solely released at the c-SMAC, the particles must be concentrated in the central region of IS. However, more evidence demonstrated that TCR$^+$ membrane particles are spread around the activated T cell periphery on the matrix-coated surface. Further, TCRs were clearly localized through the microvilli stems and tips,

which are enriched with ezrin and F-actin, and were released through long rod-shaped fragments. In addition, it is now generally accepted that microvilli play an important role in sensing, particularly at the interface with the APCs[12,31,37].

Of great interest is that most studies related to exosomes or microvesicles have focused only on the effect of these small vesicles on the target cells, tissues, or microenvironments, but none to our knowledge has yet explored whether the release of EVs has an autocrine regulatory function. From this perspective, it is striking that TISs contain, in addition to TCRs, a large number of membrane microvilli or lipid raft proteins essential for T cell life. Hence, there are two questions, viz., is the molting process unique to T cells, or do other cells also undergo molting during their lifespan? and if molting occurs, what biological or pathological phenomenon is it related to? Scientists in cancer research have believed for several years that the secretion of EVs could be one of the primary mechanisms by which cancer stem cells interact with other tumor and non-tumor cells[38]. However, no one has evaluated whether EV release is a self-regulatory mechanism by which cancer cells permanently grow. In this regard, it is interesting to note that interfering with EV biogenesis and/or release inhibits cancer cell survival and growth[38]. Furthermore, extensive evidence suggest that EVs are involved in the metabolic switch occurring in cancer and tumor stroma cells. If molting is a general phenomenon for cell proliferation, it may also be important in stem cell research. There is evidence that stem cells use plasma membrane protrusions as platforms for EV shedding[39].

In general, organisms will become exhausted and eventually die if they lose components essential for sustaining their life cycle. However, a natural phenomenon for which such a paradox exists is in molting, shedding, or ecdysis, in which a creature casts off a part of its body to grow. In this perspective, an important question is how membrane shedding was molecularly linked to enhance metabolic reprogramming for cell proliferation and survival. Since microvilli are enriched with membrane essential lipids, such as cholesterols, glycerolipids, and FAs, which are necessary for T cell metabolism and subsequent activation and proliferation, the loss of such essential lipids via TISs is likely to promote the de novo synthesis of gene clusters for lipid metabolism. This pathway may be linked to trigger Notch1 signaling, which is known to be involved in T cell proliferation and survival[40]. Additionally, no energy burden in using ubiquitin machinery to clear unnecessary receptors or signaling molecules may also be linked to Notch1 activation, thereby inducing the downstream cell cycle regulator c-Myc[40]. In addition to membrane components, TISs contain significant amounts of mitochondrial proteins. The lack of energy-related mitochondrial proteins may promote the synthesis of genes related to the OXPHOS pathway. Overall, the present results of LC-MS/MS-based lipidomics and proteomics and transcriptome analysis strongly suggest that T cells release molecules that are essential for antigen recognition (TCR and its complex), energy metabolism (glycolysis, TCA cycle, and ATP binding proteins), and membrane

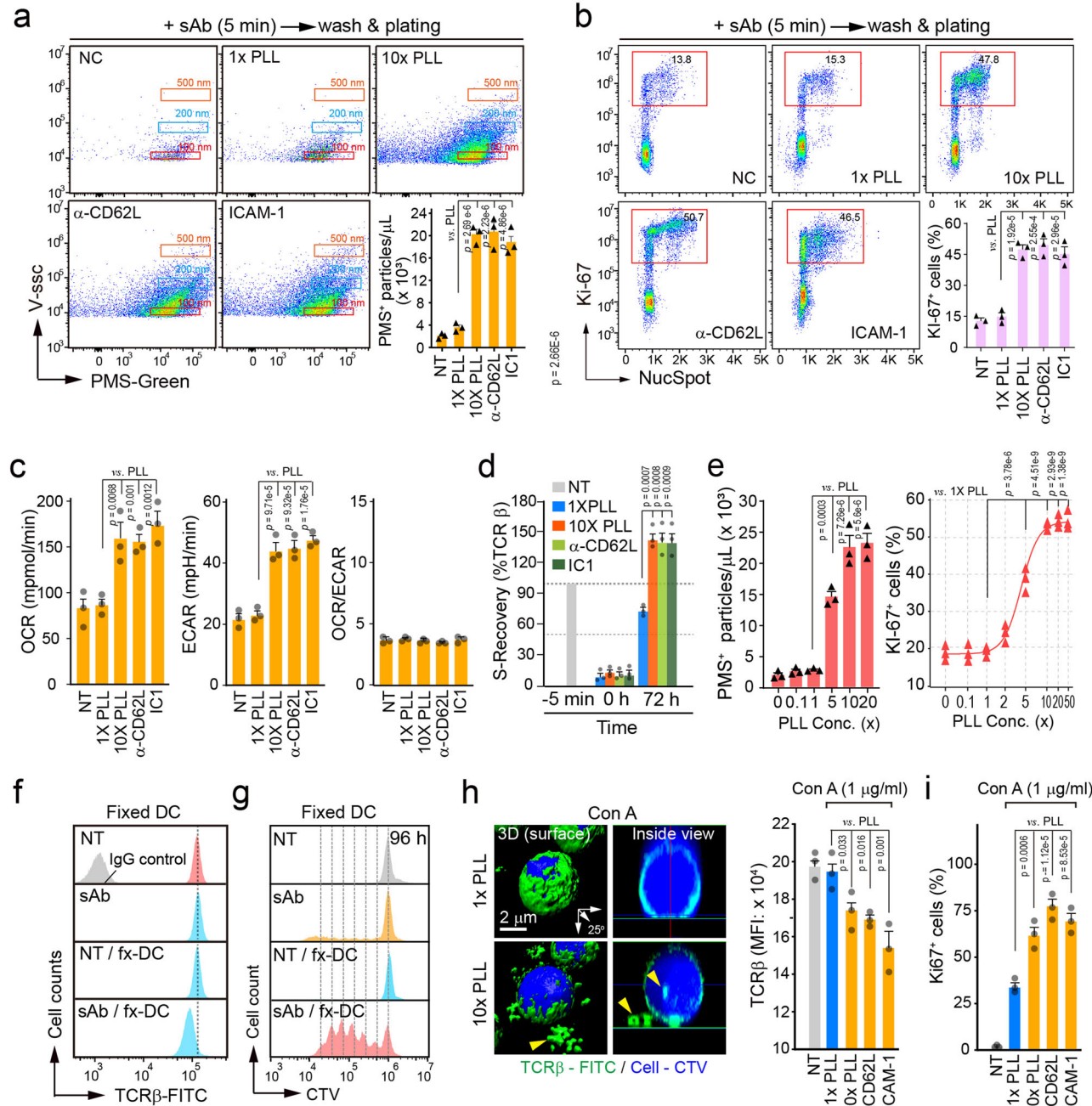

**Fig. 5 | Molting of membrane microvilli is adhesion-dependent and critical for T cell clonal expansion. a** PMS-labeled naive CD3+ T cells were stimulated for 5 min with sAb, washed, and placed on the indicated adhesion matrix-coated plates for 3 h. The number of PMS+ particles at each condition was quantified. **b–e** Naive CD3+ T cells pre-stained with anti-TCRβ-Alexa 488 were stimulated as in (**a**) and cultured for the indicated time. Ki-67+ cells at 48 h (**b**), OCR and ECAR at 24 h (**c**), and the surface recovery of TCRβ at 72 h (**d**) were determined. **e** Comparison between the released PMS+ particles and cell proliferation in conditions with various concentrations of PLL. **f–i** CTV-labeled/TCRβ-FITC-stained naive CD3+ T cells were stimulated for 5 min as in (**a**), and then incubated for 3 h with fixed DCs (**f, g**) or were stimulated with ConA for 3 h on the indicated adhesion matrices (**h, i**). TCRβ-FITC intensity, cell proliferation (CTV), and Ki-67+ cell population were measured by flow cytometry (**f, g,** and **i**). Distribution of TCRβ-FITC was observed under AiryScan confocal microscopy (**h**). Yellow arrowheads indicate released or internalized TCRβ. Data represent the mean of three independent experiments ± SEM. Statistics were performed using one-way ANOVA with post hoc Tukey's multiple comparisons test. Source data are provided as a Source Data file.

components (FAs and cholesterols), which paradoxically promote T cell proliferation for clonal expansion.

Approximately 20 years earlier, the Schoenberger group reported that naive cytotoxic T cells require a single short interaction (within ~2 h) with stimulatory APCs to initiate a program for autonomous clonal expansion and differentiation into functional effectors[41]. However, the mechanism of why a brief period was sufficient remained an enigma for a long time. In the present study, we have compiled

extensive evidence that short-term shedding of T cell membrane protrusions is a critical mechanism for initiating a program of T cell clonal expansion. We found that even ~5 min of TCR activation is sufficient, at least in vitro. Importantly, an increased number of microvilli and increased TCR expression are markers for antigen-experienced or memory T cells. Further research is underway to explore whether either membrane molting or TCR internalization is coupled with the conversion to the memory phenotype.

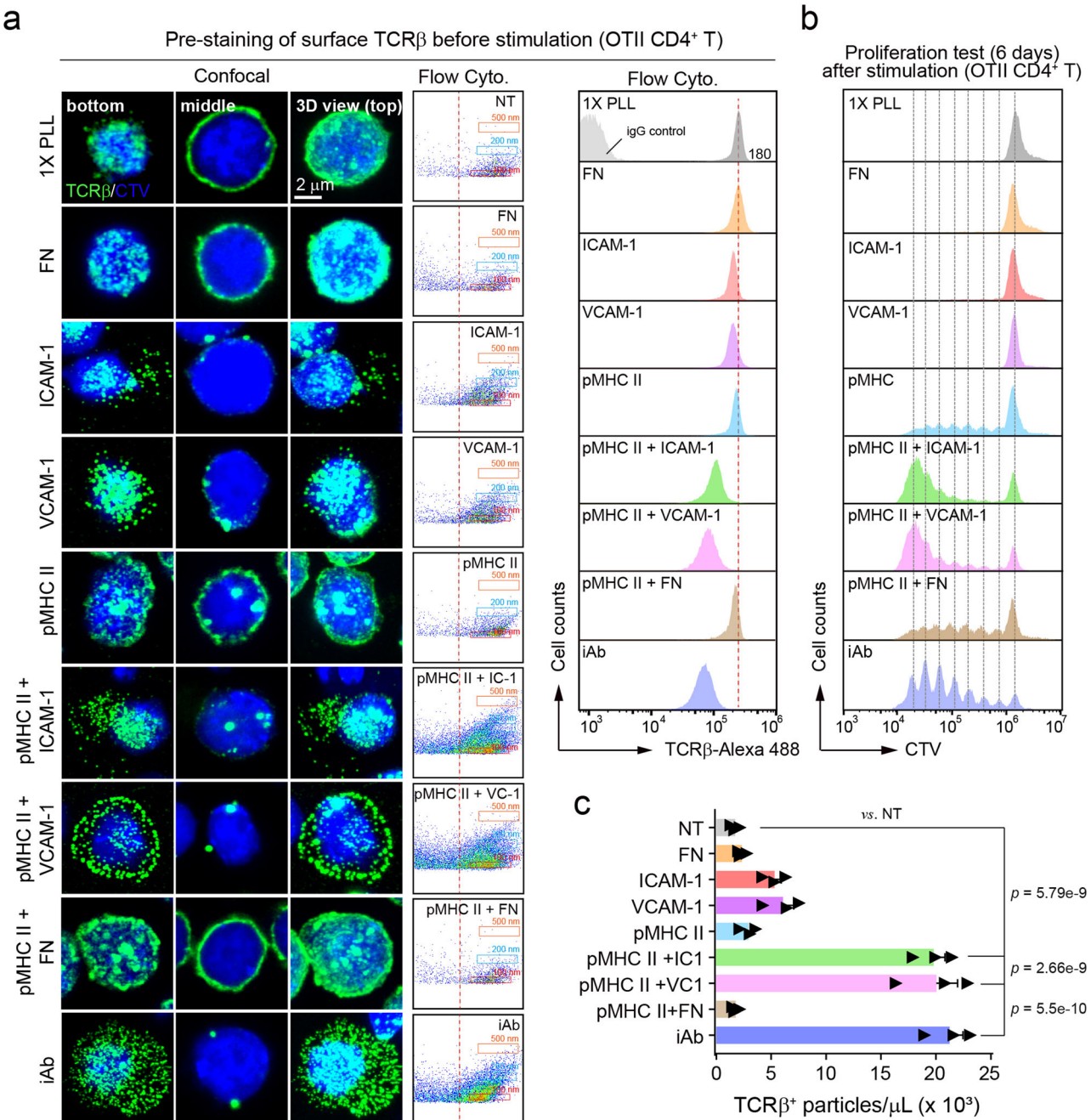

**Fig. 6 | The amount of TCR⁺TIS release is correlated with the enhanced T cell proliferation under physiological setting. a** CTV-labeled/ anti-TCRβ-Alexa 488-stained OTII CD4⁺ T cells were plated on pOVA$_{323-339}$/I·A$^b$ (2 μg/ml) plus various adhesion matrices including ICAM-1, VCAM-1, and fibronectin (FN) (each 10 μg/ml) for 3 h. The release of TCRβ⁺ particles (TISs) was determined by confocal microscopy (**a**, left) and flow cytometry (**a**, middle). Surface TCRβ exression was also analyzed (**a**, right). **b** The cells from (**a**) were cultured for 6 d with fresh media and the cell division was measured. Source data are provided as a Source Data file. **c** The number of TCRβ⁺ particles at each condition (a) was quantified by CytoFLEX. Statistics was performed using one-way ANOVA post hoc Tukey's multiple comparisons test. Results are representative of three independent experiments ± SEM. *p* value was indicated vs. NT.

## Methods
### Antibodies and reagents
Antibodies for GFP (ab290), Annexin V/propidium iodide (PI) stain kit, and fatty acid oxidation assay kit (ab118183) were purchased from Abcam (Cambridge, MA, UK). Antibodies for CD3 ζ (sc1239) and flot1 (sc-74566) were purchased from Santa Cruz Biotechnology (Dallas, TX, USA). Antibodies against β-actin (4967 L), phospho-Zap70 (2701 S), Notch1 (3608 S), phospho-PKCδ/θ (9376 S), phospho-p44/42 MAPK (Erk1/2, 4376 S), phospho-p38 MAPK (9215 S), and phospho-Akt (4058 S), in addition to anti-rabbit IgG-HRP (7074 S) and anti-mouse

IgG-HRP (7076 S), were purchased from Cell Signaling Technology (Danvers, MA, USA). Anti-TCRβ (H57-597) was purchased from Bio-X-Cell (West Lebanon, NH, USA). Anti-LFA-1 antibody (M17/4) was purchased from BioLegend (San Diego, CA, USA). Fluorescence-conjugated CD3e, CD62L, CXCR4, CD45, CD25, CD11c, CD69, CD4, and CD8 were purchased from eBioscience (San Diego, CA, USA). Flow cytometry submicron particle size reference kit (F13839), Fab preparation kit, CellMask™ Green-, Orange- Plasma Membrane Stain (CMS), CellTrace™ Violet (CTV) cell proliferation kit, CMRA-Orange, Cholera Toxin Subunit B (Recombinant), Pierce™ FITC Antibody

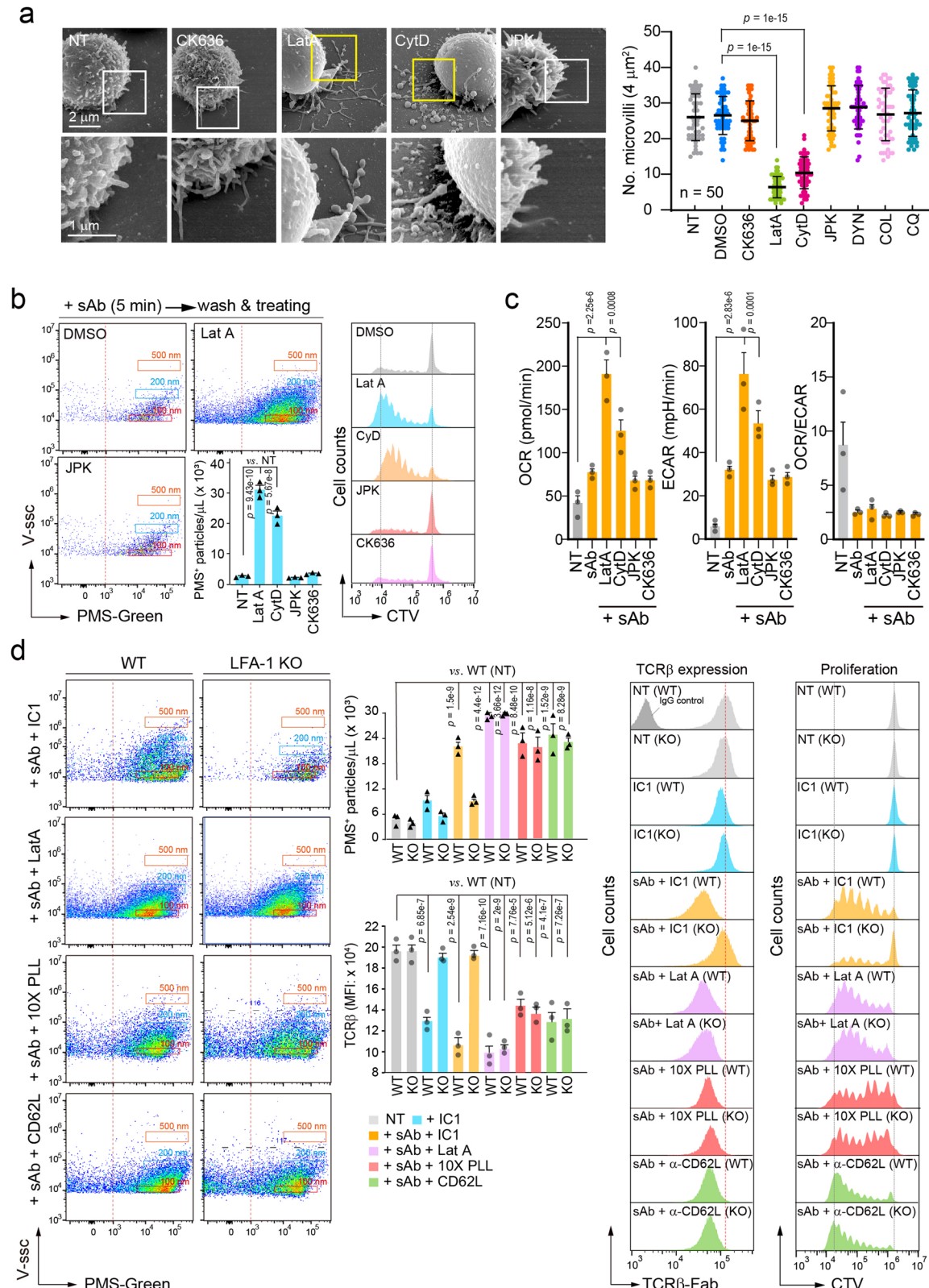

Labeling Kit, Alexa Fluor™ 488, Alexa Fluor™ 594 or APC Protein Labeling Kit, and NucSpot Far-Red were purchased from Thermo Fisher Scientific (Waltham, MA, USA). Anti-human CD28 antibody, recombinant mouse ICAM-1/CD54 His-tag, and recombinant mouse VCAM-1/CD106 Fc were purchased from R&D Systems (Minneapolis, MN, USA). Hybridoma cell lines for mouse anti-CD3 (145-2C11; CRL-1975), mouse anti-CD28 (PV1; HB-12352), and anti-human CD3 (OKT3;

CRL-8001) were purchased from the American Type Culture Collection (Manassas, VA, USA). OVA peptide fragments (323–339 or 257–264) were purchased from GeneScript (San Francisco, CA, USA). Reverse transcription PCR premix and restriction enzymes were purchased from Enzynomics (Daejeon, Korea). PLL, MG132, latrunculin A (LatA), cytochalasin D (CytD), chlorpromazine (CPZ), jasplakinolide (JPK), CK636, colchicine (COL), chloroquine (CQ), dynasore (DYN),

**Fig. 7 | Surface microvilli release is directly linked to the T cell proliferation after TCR stimulation. a** Naive CD3[+] T cells were treated with the indicated inhibitors for 1 h and surface morphology was imaged by SEM. The number of microvilli was quantified using ImageJ. *p* value was represented in the figure vs. DMSO treated. **b** PMS-labeled naive CD3[+] T cells were stimulated with for 5 min sAb, washed, and treated with the indicated inhibitors for 1 h. PMS[+] particles were quantitated by CytoFLEX (**b**, *left*). Cell division (**b**, *right*) and metabolic change (**c**) were determined at day 4 and 24 h post stimulation, respectively. Data represent the mean of three independent experiments ± SEM (**a**–**c**). *p* value was represented vs. NT (**b**, **c**).

**d** Naive CD3[+] T cells from either wild-type or LFA-1-KO mice were stained with PMS-green, TCRβ-Alexa488, or CTV and stimulated as in (**b**). The number of PMS[+] particles and surface TCRβ were analyzed (*Left*). *p*-value was represented vs. wild-type NT. Representative histogram of TCRβ expression after stimulation and cell division at day 4 were presented (*right*). Data represent the mean of three independent experiments ± SEM. *p* value was represented *vs.* wild type NT. Statistics were performed using one-way ANOVA post hoc Tukey's multiple comparisons test (**a**–**d**). Source data are provided as a Source Data file.

fibronectin, concanavalin A (ConA), and L-carnitine were purchased from Sigma (St. Louis, MO, USA). Vps4a- and Vps4b-targeting and scrambled control siRNAs were purchased as a pool of four si-RNA duplexes from Dharmacon (Lafayette, CO, USA). *Flot1* mouse siRNA was purchased from OriGene (Rockville, MD, USA). MHC I-A[b] and H-2K[b] were provided by the NIH Tetramer Core Facility (Atlanta, GA, USA). Palmitate-BSA was purchased from Cayman Chemical (Ann Arbor, MI, USA).

### Cells
Jurkat T cells (ATCC, TIB-152) or COS-7 cells (ATCC, CRL-1651) expressing Vstm5 fused with GFP (V5G) have been described previously[13]. Naive CD3[+], CD4[+], or CD8[+] T cells from C57BL/6, *OTII TCR*, or *OTI TCR* C57BL/6 mice, respectively, were purified from mouse spleens and lymph nodes by negative selection using EasySep™ isolation Kit (Stemcell Technologies, Vancouver, Canada, cat# 19851, cat# 19852, cat # 19853) or MojoSort™ Magnetic Cell Separation (BioLegend, cat# 480033) according to the manufacturer's instructions. Briefly, to obtain single cell suspensions, lymph nodes and spleens were harvested, placed in a tissue culture plate (Corning, NY, USA), crushed with a plunger, and filtered through a 40 μm cell strainer (Corning, NY, USA) to remove aggregates. Cells were then centrifuged at $2000 \times g$ for 5 min and red blood cells (RBCs) were lysed using RBC-lysis buffer (eBioscience, San Diego, CA, USA). For the generation of OTI or OTII T blast cells, CD4 or CD8 T cells from the spleen and lymph nodes of *OTI* or *OTII TCR* mice were stimulated with 2 μg/mL of plate-bound anti-CD3/CD28 antibodies in the presence of rIL-2 (100 U/mL) for 2 days. For the establishment of B-cell blasts, CD19[+] cells were purified from C57BL/6 wild-type mice using the EasySep magnetic separation system (Stemcell Technologies) and were activated with lipopolysaccharide (10 μg/mL) for 3 days in complete RPMI 1640 medium. For the isolation of bone marrow–derived dendritic cells (BMDCs), bone marrow was flushed from the femur and tibia bones, and $5 \times 10^6$ cells were cultured in 10 mL of RPMI 1640 medium supplemented with 20 ng/mL recombinant murine granulocyte-macrophage colony-stimulating factor (GM-CSF) for 9 days. GM-CSF was added at every 3 days. To generate fixed DCs, $1 \times 10^6$ DCs were placed on a fibronectin-coated plate (10 μg/mL) for 2 h, fixed with 0.4% paraformaldehyde for 60 s at room temperature, washed three times with 1× PBS, and stored at 4 °C until use.

### Animals
C57BL/6 wild-type, *OTI*, or *OTII* transgenic mice (C57BL/6 background) were purchased from Jackson Laboratory (Bar Harbor, ME, USA). In general, sex-matched, 8-week-old mice were used. LFA-1-knockout mice were provided by Dr. Minsoo Kim (University of Rochester, NY, USA). All mice were bred under specific pathogen-free conditions on a 12-h light/dark cycle at the Laboratory Animal Resource Center located in Gwangju Institute of Science and Technology. During the experiments, the mice were euthanized according to the Association for Assessment and Accreditation of Laboratory Animal Care International (AAALAC) guidelines. All experimental methods and protocols were approved by the Institutional Animal

Care and Use Committee of the School of Life Sciences, Gwangju Institute of Science and Technology, and carried out in accordance with their approved guidelines (IACUC GIST-2020-107, GIST-2020-026, GIST-2017-103, and GIST-2018-053).

### T cell stimulation
Naive CD3[+] T cells were purified from mouse spleens and lymph nodes by negative selection using EasySep™ isolation Kit (Cat # 19851) and generally activated by incubation on plate-immobilized anti-CD3 (2–10 μg/mL)/CD28 (2 μg/mL) (immobilized conditions, iAb) or by treatment with soluble anti-CD3 (2–10 μg/mL)/CD28 (2 μg/mL) followed by anti-hamster IgG as a secondary antibody (soluble conditions, sAb)s for 5 min, 30 min, or 3 h. Cells were then harvested, washed, and incubated for further 48–72 h. For the treatment of inhibitors targeting endocytosis or degradation, CD3[+] T cells were treated with chlorpromazine (20 μM), MG132 (20 μM), CQ (20 μM), or PP2 (10 μM) for 30 min at 37 °C, and then the cells were pre-stained with anti-TCRβ-FITC (anti-TCRβ-Alexa488 or anti-TCRβ-APC) for 1 h at 4 °C. Cells were then washed and stimulated with iAb or sAb for 3 h at 37 °C $CO_2$ incubator. To examine the effects of actin modulators, CD3[+] T were stimulated with sAb for 5 min, washed, and treated with various inhibitors (CK636, 100 μM; JPK, 100 nM; Lat A, 237 nM; Cyt D, 200 nM; DYN, 80 μM; COL, 10 μM; CQ, 20 μM) for 1 h. Cells were then washed and cultured for the indicated time periods. In some case, OTI naive CD8[+] T cells were stimulated for 1 h on a lipid bilayer presenting pOVA$_{257–265}$/H-2K[b]/ICAM-1 in the presence of various actin modulators.

For the different stimulations, we used ConA (1 μg/mL), PMA/ionomycin (200 nM/1 μM), PLL (0.1× – 20×; 1×, $5 \times 10^{-4}$%), recombinant ICAM-1 (10 μg/mL), VCAM-1 (10 μg/mL), fibronectin (10 μg/mL), anti-CD62L antibody (10 μg/mL), pMHC I-A[b] or H-2K[b] (2 μg/mL-coated plate), or fixed DCs. For the lipid bilayer, the density of ICAM-1 was 50–500 molecules/μm[2], unless otherwise noted, and a density of 200 molecules/μm[2] was used. OTII CD4[+] or OTI CD8[+] T cells were stimulated on lipid bilayers presenting 1 μg/mL of OVA$_{323-339}$/I-A[b] or $_{257-264}$ H-2K[b] with or without ICAM-1. In some experiments, OTII CD4[+] or OTI CD8[+] T cells were activated by co-cultured with pOVA$_{323–339}$ or $_{257–264}$-pulsed B cells or DCs. To generate effector T cells, naïve CD3 + T cells were stimulated with 2 μg/mL of plate-bound anti-CD3/CD28 antibodies in the presence of rIL-2 (100 U/mL) for 2 days, detached, and further cultured for 3 days.

### Cell transfection and viral infection
For retroviral transduction, mouse CD4[+] T cells were isolated from lymph node and spleen obtained from *OTII* TCR transgenic mice using CD4[+] T cell EasySep™ isolation Kit (Cat # 19852) were incubated in 2 μg/mL of anti-CD3/28-coated culture plates with 100 U/mL rIL-2 for 48 h. Retroviral particles were generated by transfection with Vstm5_GFP_pMSCV (V5G) or TCRζ_GFP_pMSCV along with the pCL-Eco packaging vector using Lipofectamine 2000 (Invitrogen, Carlsbad, CA, USA). After 48 h, viral supernatants were harvested, mixed with $1 \times 10^6$ mouse T cells, placed on 20 μg/mL RetroNectin (Clontech, Mountain View, CA, USA)-coated 12-well plates, and spin-infected at $2000 \times g$ for 90 min at 25 °C with rIL-2 (100 U/mL). The transduced T cells were maintained with fresh mouse T media with rIL-2 and expanded for

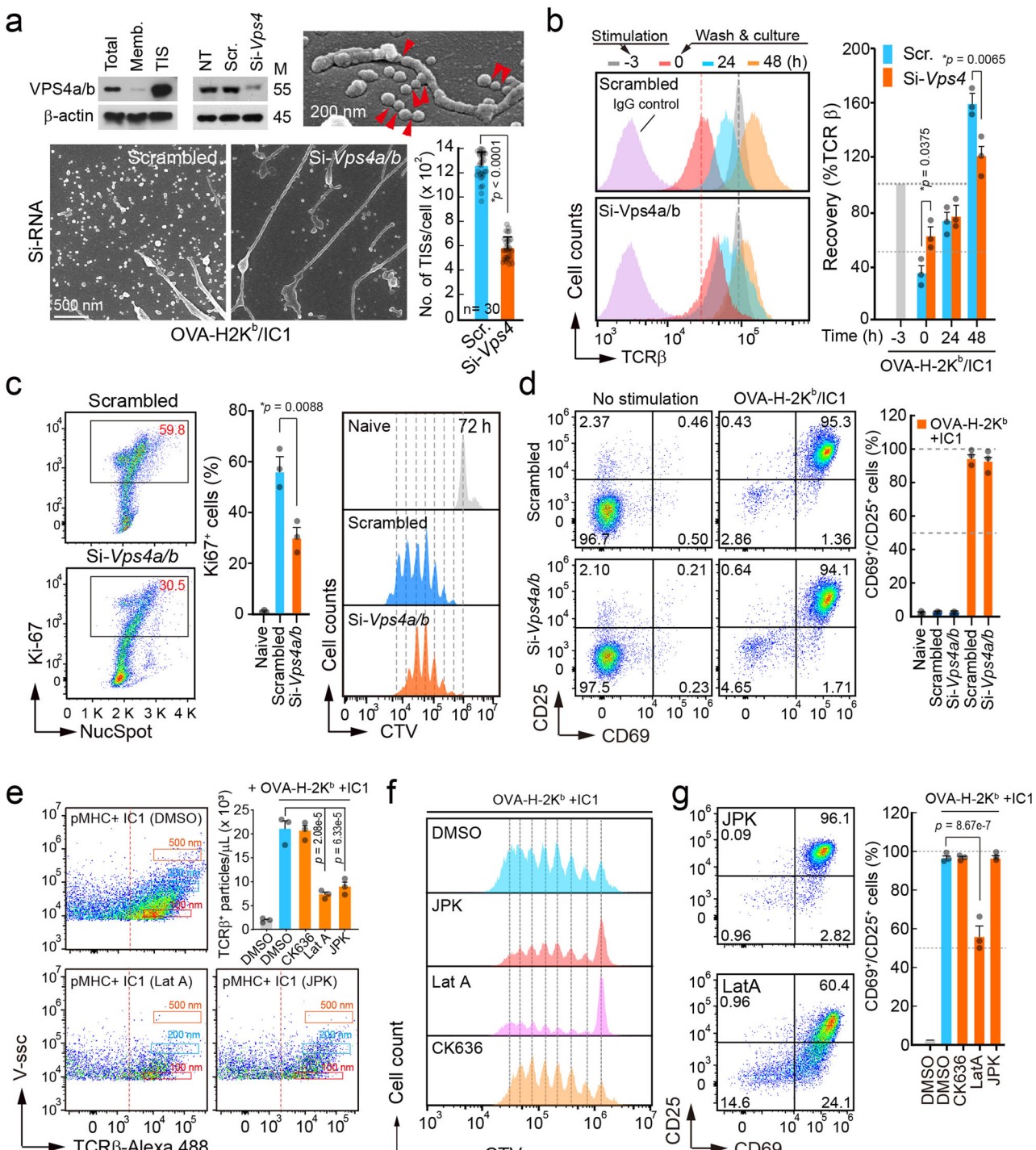

**Fig. 8 | Perturbation of microvilli shedding affects TCR recovery and T cell proliferation. a** OTI CD8[+] T blasts were transfected with scrambled or siRNA targeting Vps4a/b. At 24 h post transfection, knock-down efficiency was confirmed by western blot and cells were stimulated on a lipid bilayer presenting pOVA$_{257-265}$/H-2K$^b$/ICAM-1 for 3 h. TISs generation was observed using SEM. Results are representative of three independent experiments. **b–d** Naïve OTI CD8[+] T cells transfected with siRNA were stained with CTV and stimulated as in (**a**). Surface TCRβ recovery was determined by anti-TCRβ-FITC at the indicated time points (**b**), proliferating populations by Ki-67/NucSpot staining (at 48 h), cell division (**c**), and the expression of CD25/CD69 (at 24 h) were measured (**d**). Results are representative of three independent experiments ± SEM. p value were indicated in the figure vs. Scrambled. **e–g** OTI naïve CD8[+] T cells were pre-stained

with TCRβ-Alexa 488 or CTV and stimulated on a lipid bilayer presenting pOVA$_{257-265}$/H-2K$^b$/ICAM-1 in the presence of CK636 (100 μM), JPK (100 nM), or Lat A (237 nM) for 1 h. The number of TCRβ$^+$ particles was quantified using CytoFLEX (**e**). Results are representative of three independent experiments ± SEM. p value was represented vs. DMSO-treated OTI cells stimulated with OVA-H-2K$^b$ + IC1. Cell division and the expression of CD25/CD69 were measured at day 4 (**f**) and at 24 h (**g**) post stimulation, respectively. Results are representative of three independent experiments ± SEM. p value was represented in the figure vs. DMSO-treated OTI cells stimulated with OVA-H-2K$^b$ + IC1. Statistics was performed using unpaired two-tailed t test (**a, c**), one-way ANOVA with post hoc Tukey's multiple comparisons test (**b, e, g**). Source data are provided as a Source Data file.

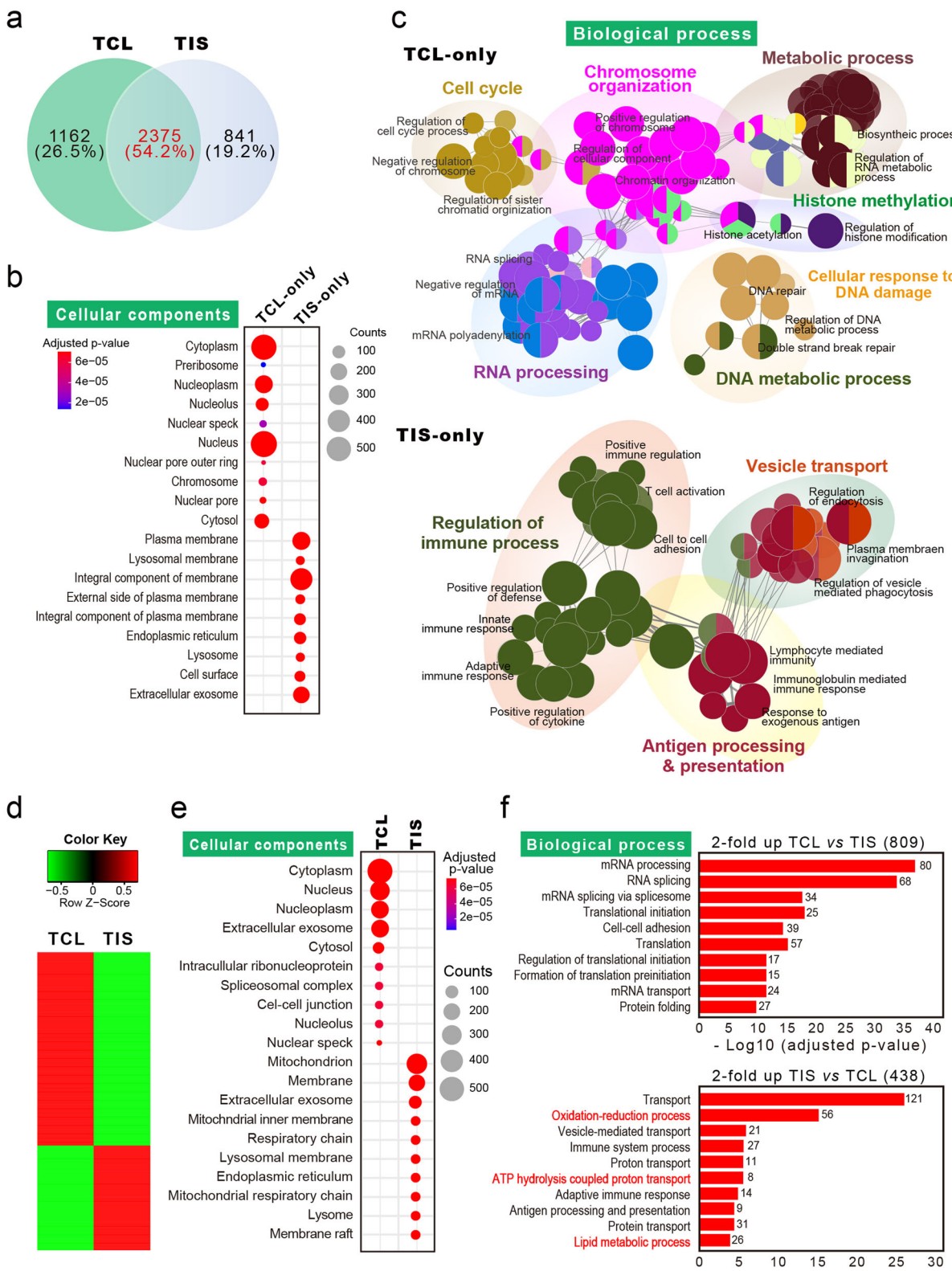

3 days. Knockdown of Vps4a/b (or scrambled, 10 nM each) was performed in naïve or activated of CD8⁺ T cells from *OTI* TCR transgenic mice using Amaxa Nucleofector II and the Mouse T cell Nucleofector Kit (Lonza, Basel, Switzerland) according to the manufacturer's instructions. For transient transfection, COS-7 or Jurkat T cells were transfected with V5G_N1 using Lipofectamine 2000 or Amaxa Nucleofector II, respectively.

**Reverse transcription PCR and real-time quantitative (q) PCR**
Total RNA was isolated from cells using the TRI reagent (Molecular Research Center, Cincinnati, OH, USA) and reverse-transcribed using RT-Premix (Intron Biotechnology). PCR was performed with the following primers (the respective forward and reverse pairs are indicated): mouse *TCR α*, 5′- CATCCAGAACCCAGAACCTGC −3′ and 5′- GGG AGTCAAAGTCGGTGAACA −3′; and mouse *GAPDH*, 5′-GCACAGTCA

**Fig. 9 | TISs contain external membrane components involved in immune and metabolic processes. a** Venn diagram showing the overlap of proteins between TCL and TIS. **b** Analysis of the GO cellular components for the proteins exclusively identified in TIS (TIS-only) or TCL (TCL-only) using DAVID 6.8 (https://david.ncifcrf.gov/). **c** Overrepresented GO network for the proteins of TIS-only and TCL-only using Cytoscape V3.6.1 (http://cytoscape.org/) with the ClueGO V2.5.7 plugin. Each node represents a GO biological process, and the size of the node reflects the enrichment significance. Node color represents the class that they belong, and mixed color indicates multiple classes. Only significant interactions ($p < 0.05$) and grouped terms are shown. The two-sided hypergeometric test yielded the enrichment for GO, and Bonferroni step-down correction for multiple testing controlled the $p$ values. **d** Heatmap showing the relative expression pattern of 1247 differentially expressed proteins in TCL and TIS. Proteins included in these analyses met a cutoff of fold change >2. **e, f** Analysis of GO cellular component and GO biological process for the proteins in (**d**) using DAVID bioinformatics resources 6.8. Benjamini−Hochberg adjustment was used for multiple test correction.

AGGCCGAGAAT-3′ and 5′-GCCTTCTCCATGGT GGTGAA-3′. Amplification was performed in a StepOne real-time PCR system (Applied Biosystems, Norwalk, CT, USA) for continuous fluorescence detection in a total volume of 10 μL of cDNA/control and gene-specific primers using SYBR Premix Ex Taq (TaKaRa Bio). The mRNA levels of the target genes were normalized relative to those of *Gapdh* using the following formula: relative mRNA expression = $2^{-(\Delta Ct\ of\ target\ gene\ -\ \Delta Ct\ of\ GAPDH)}$, where Ct is the threshold cycle value. In each sample, the expression of the analyzed gene was normalized to that of *Gapdh* and described as the mRNA level relative to *Gapdh*.

### Flow cytometry analysis

To quantitate total TCRs under various types of stimulation, cells were stained with TCRβ (H57-597 Fab-FITC, -Alexa488, -Alexa594, or -APC) antibody for 1 h at 4 °C before or after stimulation, and the mean fluorescence intensity (MFI) was measured by flow cytometry. In general, cells were stained with the indicated antibodies for 1 h in 50 μL of FACS buffer (2% FBS in PBS) at 4 °C and washed three times with 1× PBS before analysis.

For quantification of the percentage (%) of TCR recovery on surface, the MFI of TCR expression in T cells at −3 h (resting condition) was set as 100%. The recovery index was presented as the MFI at each time point divided by the MFI at −3 h. For the analysis of apoptosis, cells were stained with Annexin V and PI according to the manufacturer's instructions. Data were acquired using a FACS Canto flow cytometer (Becton Dickinson, Franklin Lakes, NJ, USA) and analyzed using the FlowJo software (TreeStar, Inc., Ashland, OR, USA). For the proliferation assay, naive T cells were stained with CTV and activated as described in the T cell stimulation section; cells were then washed, further cultured with fresh media for the indicated time periods in the presence of rIL-2, and analyzed at 72 h (CD8+ T) or day 6 (CD4+ T) after stimulation. In another experiment, cells were permeabilized at 48 h post stimulation using Cytofix/Cytoperm (Invitrogen) and stained with a mixture of anti-Ki-67 antibody and NucSpot Far-Red for 15 min at room temperature in the dark. The samples were washed with PBS and analyzed by flow cytometry. The percentage of proliferative populations was acquired from the gate in a CTV+ population. All information on antibodies used in flow cytometric analysis are provided in Supplementary Table 1 in Supplementary information.

### In vivo experimental setting for antigen-dependent TCRβ release

Wild-type recipient mice were intraperitoneally administered with either 100 μg of pOVA 257-264 or 323-339, respectively. At 48 h post injection, naïve OTII CD4+ T cells ($1 \times 10^7$) pre-stained with anti-TCRβ-Alexa 488 along with CTV were intravenously injected to recipient mice. After 4 h, total cells from the spleen and lymph nodes were isolated and RBCs were lysed. TCRβ intensity was then analyzed after the gating the CTV+ T cells.

### Confocal and TIRF microscopy

To investigate the distribution of TCRβ under the different stimulations (iAb and sAb), naive CD3+ T cells were stained with anti-TCRβ-Alexa 594 and PMS-Green for 1 h at 4 °C and stimulated with iAb or sAb for 2 h. In some experiments, cells labeled with PMS-Orange and CTV were further stained with anti-TCRβ-FITC for 1 h at 4 °C and then

treated with various inhibitors for 30 min at 37 °C. Cells were then washed and stimulated with iAb or sAb for 3 h. For imaging of sAb stimulated samples, cells were placed on 1 × PLL coated confocal dish for 5 min. To determine the release of TCRβ+ particles on DCs during synapse formation, CTV-labeled OTII CD4+ T cells were stained with anti-TCRβ-FITC and incubated with DCs (CMRA-Orange) in the presence or absence of pOVA323−339 for 2 h. In some experiments, T cells were pretreated with anti-LFA-1 blocking antibody (M17/4, 10 μg/mL) for 30 min. For intracellular staining, cells were fixed with 4% paraformaldehyde, permeabilized with 0.1% Triton X-100 in PBS, blocked with 1% BSA/PBS for 1 h, rinsed with PBS, and incubated overnight with anti-ezrin antibody at 4 °C. Secondary antibodies were added after washing and incubated in the dark for 1 h at room temperature. For the imaging of TISs generations, OTI CD8+ T cells were stained with anti-TCRβ-FITC and stimulated on the lipid bilayer presenting pOVA257−264/H-2Kb/with a different dose of ICAM-1. For actin staining, the cells were fixed with 4% paraformaldehyde, permeabilized with 0.1% Triton X-100 in PBS, and incubated with TRITC-phalloidin in PBS for 30 min at room temperature in the dark. All samples were then imaged using a 100× NA 1.40 oil immersion objective lens and an FV1000 laser scanning confocal microscope (Olympus, Japan) or a Plan-Apochromat 100 ×/1.46 NA oil immersion objective lens and Zeiss LSM 880 – Airyscan (Carl Zeiss, Germany).

To monitor the TISs generation during T cell kynapse, V5G+-transduced OTII CD4+ T blasts were placed on the lipid bilayer presenting pOVA323−339/I-Ab/ICAM-1 and immediately imaged for long time periods (~2 h) by TIRFM (IX-81; Olympus, Tokyo, Japan) equipped with a solid-state laser (488 nm, 20 mW; Coherent, Santa Clara, CA, USA).

### Electron microscopy

For transmission electron microscopy, isolated TISs were fixed in suspension at 37 °C in a 5% $CO_2$ incubator and TISs containing the sample (1 mL) were added to 9 mL of fixative (2.2% glutaraldehyde in 100 mM $NaPO_4$ [pH 7.4]) for 2 h. Samples were postfixed in 1% osmium tetroxide, stained *en bloc* with 0.5% uranyl acetate in water, dehydrated in a graded ethanol series, embedded, and thinly sectioned. The sections were stained with 2% uranyl acetate in methanol for 20 min, followed by lead citrate for 5 min, and then observed under a Tecnai G2 electron microscope (FEI, Hillsboro, OR, USA) at 120 kV under low-dose conditions. For SEM, cells were fixed with a 2.5% glutaraldehyde solution and osmium tetroxide for 2 h. Samples were then dehydrated in a graded ethanol series for 30 min, dried, prepared by sputter coating with 1–2-nm gold–palladium, and analyzed using a field-emission SEM (Hitachi, Tokyo, Japan).

### SEM analysis of microvilli number and length

All SEM imaging data, including microvilli around one single cell, were analyzed using the ImageJ software and FiloQuant plugin. For quantification of the number and length of microvilli inner area of the cells, the following six major steps were applied: (1) background subtraction, (2) sharpening and finding edges, (3) smoothing and converting to black and white, (4) closing and filling holes, (5) denoizing and segmenting, and (6) filtering and measuring. The number and length of cell-edge filopodia were automatically analyzed using the ImageJ software and FiloQuant plugin. A distribution scatter plot of the

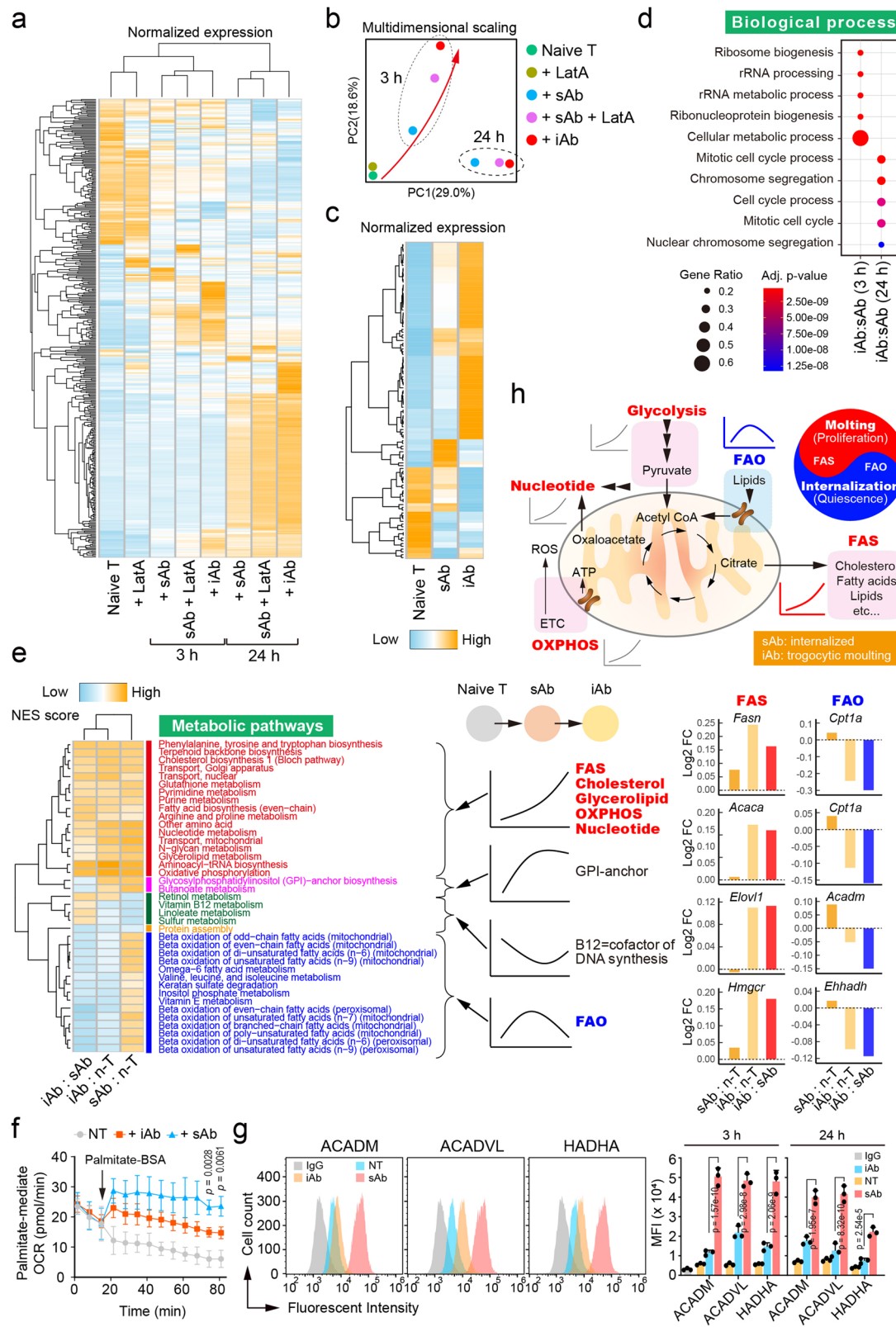

numbers of microvilli and TCR clusters was created using the Graph-Pad Prism 8 software (San Diego, CA, USA).

**TIS isolation and purification**

The purification protocol of TISs was modified according to a previously published study[13]. A total of 20 million cells were resuspended in serum-free RPMI 1640 medium and incubated on anti-CD3/CD28

antibody-coated plates for 2 h at 37 °C. Cultured media were aspirated, cells were harvested with 5 ml of PBS by gentle pipetting. Cells were then removed by centrifugations at $2000 \times g$ for 10 min and supernatants were saved. TISs on culture dishes were dissociated by gentle pipetting and scraping in the presence of cold PBS containing 10 mM EDTA. All supernatants were mixed and subjected to two successive centrifugations at $2000 \times g$ for 10 min and $2500 \times g$ for 10 min to

**Fig. 10 | Transcriptome analysis of naive T and sAb-, sAb+LatA-, and iAb-stimulated naive T cells. a** Coordinated changes in gene expression (Affymetrix Mouse Gene ST 2.0 microarray) in naive T (0 h) and LatA-, sAb-, sAb+LatA-, and iAb-stimulated T cells at 3 and 24 h. Genes on the heatmap were selected based on fold changes from naive T cells >1.5 and plotted with normalized gene expressions (z-score). **b** Principal component analysis (PCA) of the transcriptome. Trajectory was changed from naive T cells to sAb-, sAb+LatA-, and iAb-stimulated T cells. The changes were time-dependent, observing different clusters by time. **c** Selected heatmap of naive T and sAb- and iAb-activated cells at 3 h. Genes on the heatmap were selected as described in (**a**). Most of the genes in the selected group exhibited a reversed pattern of expression changes between naive cells and iAb-activated cells. **d** Enriched biological pathways (Gene Ontology) of differentially expressed genes. **e** Enriched metabolic pathways of T cells. NES of metabolic pathways (R fgsea package) were calculated from the fold changes of group comparisons. Pathways were selected based on enrichment test $p$ values (<0.05). Bar graphs show the fold changes (log2) of representative genes in the FAS or FAO pathways. **f** Kinetic OCR response to palmitate. Naive CD3$^+$ T cells stimulated for 3 h with iAb or sAb, washed, and OCR was measured in the presence of 500 μM palmitate. Results are representative of three independent experiments ± SD. $p$ value was presented for iAb $vs$ sAb. **g** Levels of FAO enzymes ACADM, ACADVL, and HADHA were measured by flow cytometry in sAb- and iAb-activated cells at 3 h or 24 h. Representative histogram and the mean MFI is shown. Results are representative of three independent experiments ± SEM. **h** A schematic model based on global metabolic gene profiling and the metabolic functional changes caused by molting or TCR internalization. Statistics was performed using one-way ANOVA with post hoc Tukey's multiple comparisons test. FAS fatty acid synthesis, FAO fatty acid oxidation, GPI glycosylphosphatidylinositol, and OXPHOS oxidative phosphorylation, ACADM, medium-chain acyl-CoA dehydrogenase, ACADVL very long chain acyl-CoA dehydrogenase, HADHA trifunctional multi enzyme complex subunit alpha. Source data are provided as a Source Data file.

completely eliminate cells and debris. TISs in PBS were reloaded onto a sucrose gradient of 12 different sucrose concentrations from top to bottom (10–90% sucrose) and centrifuged at $100,000 \times g$ for 16 h. Fractions were then carefully collected at 2 mL each from the bottom of the tube. All fractionated samples were diluted with PBS and subjected to centrifugation at $100,000 \times g$ for 90 min to pellet the vesicles. Unless otherwise mentioned, 4–6 consecutive fractions of the 12 fractions were used for western blot, SEM, TEM, and LC-MS/MS analyses.

## Quantitation of TISs by flow cytometry

PMS-green or anti-TCRβ-Alexa488 stained T cells ($1 \times 10^6$) were stimulated as indicated and supernatant containing TISs were prepared by ultracentrifugation as described in TIS isolation and purification in *Materials and Methods* section. TISs pellet was resuspendend in 100 μL of PBS, re-stained with PMS or anti-TCRβ-Alexa488 for 1 h at 4 °C, washed with PBS, and pelleted by ultracentrifugation at $100,000 \times g$ for 1 h. TISs pellet were resuspendend in 1 mL of PBS for flow cytometric analysis. Depending on the preparation method and particular cell types, the appropriate dilution for the TISs samples before analysis was typically 1:10–1:1000 relative to the initial sample when running at a slow sample rate (10 μL/min). All experiments in this study were performed using a 3-laser CytoFLEX Flow Cytometer (Beckman Coulter, Brea, CA, USA) and operated using the CytExpert Software v2.4.0.28 (Beckman Coulter). For quantitation, the configuration was modified for VSSC detection. Briefly, the 405/10 VSSC filter was moved to the V450 channel in the wavelength division multiplexer and the detector configuration was modified in the CytExpert Software to assign the VSSC channel within the wavelength division multiplexer. The event-rate setting was set to high before initiating analyses, tightening the pulse window and thus reducing the background for small particle analyses. Finally, the trigger channel was set to VSSC-Height, and the threshold level was manually set as appropriate for small particles. The optimal threshold setting for the CytoFLEX was determined empirically using the Flow Cytometry Submicron Particle Size Reference Kit at their optimal dilution.

## Western blot analysis

Cells or TISs were lysed in ice-cold lysis buffer (50 mM Tris-HCl pH 7.4, 150 mM NaCl, 1% Triton X-100, 1× complete protease/phosphatase inhibitor cocktail) for 1 h on ice. Lysates were centrifuged at $16,000 \times g$ for 25 min at 4 °C, and the harvested supernatants were mixed with sodium dodecyl sulfate (SDS) sample buffer (100 mM Tris-HCl pH 6.8, 4% SDS, 20% glycerol, bromophenol blue) and then boiled for 5 min. Proteins were separated on 10%–12% SDS polyacrylamide gels by electrophoresis and transferred onto nitrocellulose membranes using a Trans-Blot SD Semi-Dry transfer cell (Bio-Rad, Hercules, CA, USA). Membranes were blocked with 5% skim milk for 1 h, rinsed, and incubated overnight with primary antibodies in Tris-buffered saline (TBS; 50 mM Tris-HCl pH 7.4, 150 mM NaCl) containing 0.1% Tween 20 (TBS-T) and 3% milk. The excess primary antibody was removed by washing the membrane four times in TBS-T before incubation with the peroxidase-labeled secondary antibody (0.1 μg/mL) for 2 h. Bands were visualized using a WEST-ZOL western blot detection kit and exposed to X-ray film. All uncropped blots in the main figures are provided as source data files.

## Lipid extraction and analysis

The extraction of lipids for lipidomic analysis was performed by a modified protocol based on methods previously described[42]. Total T cell lysate ($1 \times 10^7$ cells) and TISs obtained from $1 \times 10^9$ cells were lyophilised before lipid extraction and then crushed into a fine powder. Individual samples were first dried in a model Bondiro MCFD 8508 freeze dryer vacuum centrifuge (IlShinBioBase, Yangju, Korea) and the lyophilized powder was mixed with 300 μL of methanol in an ice bath and incubated for 10 min. Individual samples were added to 150 μL of a 9:1 (v/v) solution of $H_2O$ and PBS and tip-sonicated for 1 min in an ice bath. Each mixture was dissolved in 300 μL $CH_3OH$ in an ice bath and vortexed for 10 min. Following incubation or vortexing for samples, 1 mL of methyl–tert –butyl ether (MTBE) with 100 μL of MS-grade $H_2O$ was added to each sample and the mixture was vortexed for 1 h. After vortexing, the mixture was centrifuged at $3000 \times g$ for 5 min and the organic layer was transferred to a new tube. The aqueous layer was mixed with 400 μL of MTBE and vortexed for 10 min at $3000 \times g$. The resulting organic layer was added to the previously collected extract. The organic solvent of the final mixture was removed using the freeze dryer vacuum centrifuge for 3 h. To minimize the loss of lipids, the tube was wrapped with a MillWrap PTFE membrane (0.45 μm; Millipore, Bedford, MA, USA) During evaporation. Dried lipid powders were reconstituted in MeOH:$CHCl_3$ (9:1, v/v) at a concentration of 10 μg/μL and further diluted to 2 μg/μL with $CH_3OH$:$H_2O$ (9:1, v/v) for nUHPLC-ESI-MS/MS analysis. Lipid analysis was performed using two sets of nUHPLC-ESI-MS/MS systems: a Dionex Ultimate 3000 RSLCnano System with LTQ Velos ion trap mass spectrometer from Thermo Fisher Scientific (San Jose, CA, USA) for non-targeted lipid identification and a nanoACQUITY UPLC system from Waters™ (Milford, MA, USA) coupled with a TSQ Vantage triple-stage quadrupole MS system from Thermo Fisher Scientific for targeted quantification. Analytical columns were prepared in the laboratory using fused silica capillary tubes (100 μm inner diameter (I.D.), 360 μm outer diameter, 7 cm long). Prior to packing, a capillary tube exposed to flame was extended and tapered to produce a sharp needle-like tip for the self-emitter in the MS experiment. The end portion (-5 mm) of the pulled-tip column was filled with 3 μm of 100 A˚ Watchers® ODS-P C-18 particles from (Isu Industry Corp., Seoul, Korea) to form a self-assembled frit and the rest (6.5-cm) was packed with 1.7-μm XBridge® BEH resin under nitrogen gas at 1000 psi. The capillary column was connected with a capillary tube (50 μm I.D.) from the nUHPLC pump via a PEEK micro-cross (IDEX

Health & Science, Oak Harbor, WA, USA). The other two ports of the microcross were connected with a pressure capillary tube (20 μm I.D.) and an on/off switching valve to split the pump flow. For binary gradient elution, the same set of mobile phase solutions was used for both nUHPLC systems: $H_2O:CH_3CN$ (9:1, v/v) for mobile phase A and $CH_3OH:CH_3CN:IPA$ (2:2:6, v/v) for mobile phase B. Both solutions were added with a mixture of ionization modifiers, 0.05% $NH_4OH$ and 5 mM $NH_4HCO_2$, which are universal for both positive and negative ion modes of MS detection. Non-targeted lipid identification was carried out with pooled lipid extracts. For each sample group, 3 μg of lipid extract was loaded onto the analytical column from an autosampler using mobile phase A at a flow rate of 1.0 μL/min for 7 min. Gradient elution began with 60% of mobile phase B at a pump flow rate adjusted to 9 μL/min with the switching valve on for flow splitting to deliver 300 nL/min to the analytical column. Mobile phase B was ramped up to 80% for 11 min, raised to 100% for 20 min, and maintained for 7 min at 100%. Then, mobile phase B was decreased to 0% (100% A) and re-equilibrated for 10 min prior to the next run. The m/z range of a precursor MS scan was set to 300–1000 m/z for lipid detection in which range CL species were detected as $[M-2H]^{2-}$ form. The ESI voltage was set to 3.0 kV for both positive and negative ion modes. For data-dependent MS/MS experiments, 40% normalized collision energy was utilized. Targeted quantification of lipids was performed with each lipid extract of an individual animal, which was added along with 19 internal standards (ISs). For each injection, 2 μg of lipid extracts containing 1 pmol of each IS were loaded onto the same analytical column with mobile phase A for 7 min at 1 μL/min. After sample loading, the switching valve was adjusted to the split mode and the total pump flow rate was set at 16 μL/min (300 nL/min for analytical column). Gradient elution began at 60% B, was ramped to 100% for 20 min, maintained at 100% B for 4 min, and resumed at 100% A. For quantification of lipids, Selected Reaction Monitoring (SRM)-based analysis was performed by scanning the precursor ion and a selected product ion in data-dependent CID experiments in the polarity switching mode (alternating detection at positive and negative ion mode) of the triple-stage quadrupole MS system. This involved LPC, PC, LPE, PE, DG, TG, SM, Cer, HexCer, and SulfoHexCer in the positive ion mode and LPA, LPG, PG, LPI, PI, LPS, PS, MLCL, and CL in the negative ion mode. The amounts of individual lipid species were determined by calculating the corrected peak area (the ratio of the peak area of a species to the area of the IS specific to each lipid class), which is a good estimate of the pmol/mg for tissue sample. For lipid quantification, we also conducted an SRM-based analysis using product ions obtained in polarity switching mode of the triple-stage quadrupole MS system. For the quantitative normalization of lipid species between T cells and TISs, the MS peak area of individual lipid species was calculated by dividing the peak area of the internal standard. To analyze the data, we utilized Lipid Search 4.2.21 was utilized to analyze the initial LC-MS/MS files and established the necessary search parameters. *Abbreviations;* Diacylglycerol (DG), Triacylglycerol (TG), Lysophosphatidylcholine (LPC), Phosphatidylcholine (PC), Lysophosphatidylethanolamine (LPE), Phosphatidylethanolamine (PE), Lysophosphatidic acid (LPA), Phosphatidic acid (PA), Lysophosphatidylglycerol (LPG), Phosphatidylglycerol (PG), Lysophosphatidylinositol (LPI), Phosphatidylinositol (PI), Lysophosphatidylserine (LPS), Phosphatidylserine (PS), Sphingomyelin (SM), Ceramide (Cer), Cholesterol ester (ChE), monolysocardiolipin (MLCL), Cardiolipin (CL), Sulfohexosyl ceramide (SulfoHexCer), Hexosylceramide (HexCer).

### Metabolic studies

Naive CD3$^+$ T cells were placed on PLL-coated Seahorse Bioanalyzer XFp culture plates ($3 \times 10^5$ cells/well) with Seahorse XF RPMI Assay Media (RPMI Medium, pH 7.4, 103576-100; Agilent Technologies, Santa Clara, CA, USA), supplemented with 10 mM glucose (103577-100; Agilent), 1 mM pyruvate (103578-100; Agilent), and 2 mM L-glutamine

(103579-100; Agilent) for 1 h at 37 °C. Seahorse XF RPMI Assay Media supplemented with 5.5 mM glucose and 150 mM L-carnitine was used to measure fatty acid oxidation. Basal rates were taken for 24 min, and 20 μL of dynabead mouse CD3/CD28 T cell activator ($2 \times 10^6$ beads/mL; dAb, to mimic immobilized conditions) or sAb (10 μg/mL) was injected into the culture plate and stimulated rates were taken for 2 h. For screening fatty acid oxidation rate, 20 μl of Palmitate-BSA (final concentration: 500 μM) was injected. To analyze the metabolic changes under different conditions, naive CD3$^+$ T cells were stimulated with iAb, sAb, and sAb on ICAM-1-, PLL-, or CD62L-coated plates for 3 h. In some experiments, the cells were pretreated with various inhibitors (LatA, 237 nM; Cyt D, 200 nM; JPK, 100 nM; COL, 10 μM) for 1 h, washed, and stimulated with sAb for 3 h. Cells were then washed, cultured for another 21 h, and placed on PLL-coated Seahorse Bioanalyzer XFp culture plates with Seahorse XF RPMI Assay Media for 1 h at 37 °C before measurement.

### Fatty acid β-oxidation assay

Naive CD3$^+$ T cells ($3 \times 10^6$) were stimulated with iAb or sAb for 3 h and cells were then washed three times. For preparation of 24 h-activated samples, cells were washed and further cultured for 21 h after stimulation. The activated T cells were fixed with 4% PFA in PBS and permeabilized with 0.1% Triton X-100 in PBS for 15 min at RT. After rinsed three times, fixed cells were incubated with blocking buffer (1% BSA in PBS) for 15 min at RT. Next, ACADM, ACADVL, and HADHA primary antibodies in reagent buffer (0.1% BSA, 0.1% Triton X-100 in PBS) were added and incubated for 1 h at RT. After incubation, cells were rinsed three times and incubated in reagent buffer containing secondary antibody conjugated with Alexa Fluor 488 for 1 h at RT. Stained cells were washed three times with PBS and resuspended in 100 μL of PBS and MFI were measured by flow cytometry.

### Proteomic analysis by LC-MS/MS

TISs were harvested as described in the TISs isolation and purification section, and the same amount of proteins from the total lysate or TISs was subjected to SDS-PAGE on a 12% polyacrylamide gel. The gel was stained with Coomassie Brilliant Blue R-250 and fractionated according to molecular weight. Tryptic in-gel digestion was conducted according to a previously described procedure[43]. The digested peptides were extracted with an extraction solution consisting of 50 mM ammonium bicarbonate, 50% acetonitrile, and 5% trifluoroacetic acid (TFA) and then dried. For LC-MS/MS analysis, the samples were dissolved in 0.5% TFA. Tryptic peptide samples (5 μL) were separated using an Ultimate 3000 UPLC system (Dionex, Sunnyvale, CA, USA) connected to a Q Exactive Plus mass spectrometer (Thermo Scientific, Waltham, MA, USA) equipped with a nanoelectrospray ion source (Dionex). Peptides were eluted from the column and directed on a 15 cm × 75 μm i.d. Acclaim PepMap RSLC C18 reversed-phase column (Thermo Scientific) at a flow rate of 300 nL/min. Peptides were eluted by a gradient of 0%–65% acetonitrile in 0.1% formic acid for 180 min. All MS and MS/MS spectra obtained using the Q Exactive Plus Orbitrap mass spectrometer were acquired in the data-dependent top10 mode, with automatic switching between full-scan MS and MS/MS acquisition. Survey full-scan MS spectra (m/z 150–2000) were acquired in the orbitrap at a resolution of 70,000 (m/z 200) after the accumulation of ions to a $1 \times 10^6$ target value based on predictive automatic gain control from the previous full scan. The MS/MS spectra were searched with MASCOT v2.4 (Matrix Science, Inc., Boston, MA, USA) using the UniProt human database for protein identification. The MS/MS search parameters were set as follows: carbamidomethylation of cysteines, oxidation of methionines, two missed trypsin cleavages, mass tolerance for a parent ion and fragment ion within 10 ppm, and a *p* value of <0.01 of the significant thresholds. The exponentially modified protein abundance index (emPAI) was generated using MASCOT, and mol%

was calculated according to the emPAI values[44]. The MS/MS analysis was performed at least three times for each sample.

## RNA isolation and Affymetrix GeneChip microarray

For the preparation of RNA samples, $5 \times 10^6$ naive CD3[+] T cells were stimulated with Lat A (237 nM), sAb, iAb, or sAb+LatA for 3 h, washed, and harvested. For the 24-h sample, the cells were washed and further cultured for 21 h after stimulation. In some analysis, cells were stimulated with ConA, PLL, or ConA+ PLL. Total RNA was isolated using an RNeasy Plus Mini Kit (Qiagen, Hilden, NRW, Germany). cDNA was synthesized using the GeneChip Whole Transcript amplification kit purchased from Thermo Fisher Scientific (Waltham, MA, USA). cDNA was fragmented using terminal deoxynucleotidyl transferase and labeled with biotin. The labeled target DNA was hybridized for 16 h at 45 °C. Finally, the hybridization signal was calculated using the Affymetrix® GeneChip™ Command Console Software. Gene enrichment and clustering were performed using KEGG (www.genome.jp/kegg/), and a functional analysis was subjected to Gene Ontology (www.geneontology.org/).

## Expression profiling of microarray and pathway analysis

Raw data of the microarray were normalized using the robust multi-average method[45], and PCA was performed on the normalized expression profiles of all samples using the R prcomp package. Using the fold change values, we performed rank-based pathway enrichment tests from the R gProfiler (Gene Ontology-based). For in-depth metabolic pathway enrichment tests, we extracted genes belonging to metabolic pathways (i.e., metabolic subsystems) from high-quality genome-scale metabolic models of mouse (MMR version 1)[27,28] and performed pathway enrichment tests using the R fgsea package.

## Statistics

Student's $t$ test and one-way analysis (with post hoc Tukey's multiple comparisons test) of variance, corrected for all pairwise comparisons, were performed using the GraphPad Prism software version 8.1.2 (GraphPad, San Diego, CA) between two and more different groups. A $p$ value of <0.05 was considered to be statistically significant. Data are presented as mean ± SD or standard error of the mean.

## Reporting summary

Further information on research design is available in the Nature Portfolio Reporting Summary linked to this article.

# Data availability

The LC-MS/MS data generated in this study have been deposited in the PRIDE under accession code PXD041442. The microarray data have been deposited in the GEO under accession numbers GSE228653 and GSE228654. The rest of the data are available in the article, Supplementary Information, or Source Data file. Source data are provided with this paper.

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

## Acknowledgements

This work was supported by the Creative Research Initiative Program (2015R1A3A2066253); Bio-Synergy Research Project (2021M3A9C4000991); Bio & Medical Technology Development Program [2020M3A9G3080281] through National Research Foundation (NRF) grants funded by the Ministry of Science and ICT (MSIT), the Basic Science Program (2022R1A2C4002627) through National Research Foundation (NRF) grants funded by the Ministry of Education (MOE), and supported by GIST Research Institute (GRI) IBBR grant funded by the GIST (in 2021–2022), and the Joint Research Project of Institutes of Science and Technology (2021–2022), Korea.

## Author contributions

J.-S..P. conceived the study; J.-S.P., J.-H.K., N.-Y.K., W.-C.S., and K.-S.L. designed and performed the experiments; S.L. and C.-H.K. analyzed the data; H.-R.K. and C.-D.J. wrote and finalized the manuscript.

## Competing interests

The authors declare no competing interests.
