## [Peer Review File · Nature Communications]

Trogocytic molting of T cell microvilli upregulates T cell receptor surface expression and promotes clonal expansionREVIEWER COMMENTS

Reviewer #1 (expertise in trogocytosis, extracellular vesicles):

In this study, Park and co-workers address the role of T-cell microvilli shedding in T cell activation and metabolic reprogramming in response to TCR/CD3 signals. The study addresses a decades-long observation, which is that immobilized anti-CD3+anti-CD28 is more efficient than soluble anti-CD3+anti-CD28 to promote T cell activation. The authors had described TCR/CD3 complex at microvilli, and here they claim that microvilli shedding drives TCR upregulation and metabolic reprogramming towards clonal expansion.

Although of indisputable interest in the field and beyond, the study requires extensive experimentation and additional controls. As the data stand, this is a nice, but phenomenological story, which may or may not be of general interest due to the fact that non-specific adhesion triggers the microvilli removal effect. While this is important (and probably an unaddressed truth in the field), for the story to be truly relevant, one would expect this to be somewhat specific. In addition, the specific mechanisms that drive microvilli removal are cursorily examined, which is not acceptable at this stage.

Authors stimulate cells for hours or days and discuss their results as if these are physiological situations. In their system, there are no stop signals for interaction or adhesion: the use of immobilized antibodies cannot be assimilated to antigen-presenting cells. Of course, in the restricted volume of a cell culture well, cells are always subjected to the same conditions, which is not the case for circulating T cells.

MAJOR POINTS

- Perhaps the most striking piece of data is that adhesion, even non-specific, increases T cell activation (measured as CD3zeta-GFP released particles). Is this dependent on adhesive strength? Does it vary depending on the type of adhesive receptor engaged? This is an important issue as, in the lymph node, T cells are constantly surrounded by non-specific, but potentially activating, adhesive signals.

- Imaging studies are usually descriptive, with only one cell analyzed; and stimulation for 3 h cannot be considered as a short period to study changes in the membrane and cytoskeleton of T cells. Why is the TCRbeta staining in CD4 T cells from OVA+anti-LFA-1 Ab condition similar to the OVA w/o anti-LFA-1 Ab condition (but different from no peptide contact with dendritic cells; Fig 1E)? The dendritic cell does not seem to internalize any particle received by the T cell.

- Does a surface coated with ICAM-1 alone promote TCR shedding? What about VCAM-1 or fibronectin?

- A major issue with this paper is that the mechanistic connection between cell adhesion, TCR/CD3 triggering and microvilli release is barely worked out. The only mechanistic insight into this crucial issue in the story is provided in Fig. 6A. What is triggering the engagement of the abscission complex (containing Vps4)? Is it adhesion? Is it the TCR? For this part to make a real contribution, this pathway needs to be deciphered in more detail.

- Authors analyze naïve CD3 T cells, and therefore CD4 and CD8 cells indistinctly. These cells show very different behavior when referring to TCR activation, adhesion and mechanotransduction. In Fig 3D, the CD4 and CD8 receptors are barely lost upon activation with pOVA and ICAM-1; does this mean that they are not binding correctly to the MHC? Are the microvilli devoid of CD4 or CD8?

- That LatA treatment impairs T cell activation is not novel, but the reasoning the authors use to justify this result is somewhat skewed. At least with CK636, the authors can rule out the involvement of dendritic actin polymerization in the process, but the Jask and LatA experiments do not make a compelling enough argument to sustain that microvilli are critical for T cell activation by IS formation, but once TCR has been engaged, shedding is essential.

- In fact, the data reported here could be used to argue that TCR downregulation is essential for T cell activation. What happens is the authors target TCR expression transiently with, for example, siRNA, after activation? Do cells undergo a proliferative repression followed by increased

activation? Probably, microvilli in cells and TMPs recovered from surface are dissimilar in components.

- In general, if microvilli are sensory "organs" that cells use during IS formation and are shed for proper T cell activation, why do activated cells display many more microvilli after 72h? (Fig. 4B). In principle, those cells would be already activated and would have little need for additional microvilli.

- In the same vein, if those cells are re-stimulated, do they shed those microvilli as well? Or those "2nd generation" microvilli are somewhat more resistant to shedding?

- Why would TMPs contain metabolic components? Or this is just a representation of the metabolic reprogramming of activated cells? If so, shouldn't these changes be observed in total cell lysates? The rationale for performing MS, GO and the like on TMP content seems weak.

- The correlation between the type of metabolic reprogramming (FAS or FAO) and microvilli shedding is, as shown, phenomenological. The authors need to set an efficient intervention point that prevents microvilli shedding in adhered cells (not a cytoskeletal drug) and probe metabolic reprogramming in those cells.

Other comments:

- Authors should carefully check the references. References 7 and 17 are duplicated; and reference 17 is mistakenly written.

- The titles in figure legends do not correspond to the experiments showed; e.g.: in figure 2, there is no APC involved.

- The reduced resolution in Z for classical confocal microscopies does not allow proper analysis of co-localization or localization

- Fig S8B. Yellow is not included in the figure

- The use of orange and blue color in the Fig. 5 for different parameters is misleading.

- Anti-CD62L is not an irrelevant antibody (line 234) if it has so much effect on cells as observed in the release of microvilli, OCR/ECAR experiments, and on the recovery of TCR and in cell division when combined with sAb (Fig. 5).

Reviewer #2 (expertise in immune synapse, immune cell signaling):

In this report the authors address the role of TCR shedding via T cell microvilli particles (TMP) in the decrease in surface TCR that occurs following TCR engagement. In the current view, TCR downregulation is the result of ligand-dependent endocytosis, which is followed by either degradation or recycling to the T cell surface. Here the authors show that, when T cells are activated by immobilized anti-CD3 mAb, or MHC-bound peptide plus ICAM-1 on supported lipid bilayers (SLB), or by cognate dendritic cells (DC), they downregulate surface TCRs in an adhesion-dependent manner by releasing TCR+ TMPs rather than by TCR internalization. They show that TCR shedding in association with TMPs leads to an enhancement in TCR expression, metabolic upregulation (associated with fatty acid synthesis), and enhanced cell proliferation and survival. Based on these findings, they propose that TCR release through microvilli shedding triggers an intrinsic circuitry that sets the conditions for clonal expansion.

This report sheds light into the mechanisms that regulate TCR levels at the cell surface and hence the ability of the T cell to respond to cognate ligand. The impressive amount of work presented in the manuscript, which involved a variety of approaches, is of excellent quality and overall supports

the authors' conclusions. However, the results do not fully answer one of the main questions, i.e. the role of TCR release versus endocytosis in TCR downregulation. The authors have carried out all the experiments using two main activation settings (in some cases complemented by additional ones, e.g. SLBs or cognate DCs): "soluble" anti-CD3 mAb (sAb), that involves a subsequent step of cross-linking by secondary Abs, and immobilized (plate-bound) anti-CD3 mAb (iAb), in both cases in the presence of anti-CD28 mAb. The results clearly show that, starting from a different primary mechanism of TCR downregulation (endocytosis vs TMP shedding, respectively), these conditions lead to very different biological outcomes. But if, as the authors underscore, iAbs represent the condition closer to physiological stimulation compared to sAbs (as also supported by the results obtained using the SLB or DC activation settings), the data obtained using sAbs are a major confounding factor. To have clear information on the respective contribution of TCR endocytosis vs shedding in T cell activation, the two pathways should be analyzed under the same condition, namely the more physiological iAb setting. This has been done, but should be done more quantitatively. Additionally, the authors are challenging a model that has been established over decades with the contribution of many research groups. Hence the role of TCR endocytosis, as well as recycling (which has not been considered), should be addressed more in depth, experimentally and in the discussion.

Specific points

Point 1. Figure 2B. Based on size of the particles detected by quantitative flow cytometry in the culture supernatants of iAb-stimulated cells the authors propose that the 100 nm-range population corresponds to TCR+ TMPs. This should be confirmed by staining with anti-TCR antibodies as well as antibodies to microvilli markers (e.g. Vstm5)

Point 2. Figure S1. In panel A the authors used inhibitors of clathrin-dependent endocytosis as well as proteasome and lysosome-dependent degradation to address the role of these pathways, previously implicated in TCR downregulation, in the decrease in surface TCR observed in iAb-stimulated cells. Based on the lack of effect of these inhibitors, the authors conclude that TCRs are mainly released during T cell adhesion. Have the authors checked that these treatments affect the (smaller) decrease in surface TCR in sAb-activated cells, which based on their data should mainly be the endocytic one? Also, TCR internalization has been reported to be regulated in part by a clathrin-independent pathway that leads to their incorporation into an endocytic network marked by flotillins. This network is essential for TCR recycling. This pathway should also be ruled out in iAb-activated cells.

Point 3. Figure 6C. The authors show that Vps4a/b knockdown leads to attenuated metabolic reprogramming and reduced T cell proliferation. How stable is the knockdown over time? Also, Vps4, together with the ESCRT complex, is not only essential for microvilli fission but also for the generation of other membrane vesicles, including multivesicular body-derived extracellular vesicles and synaptic ectosomes. Hence these data do not provide a final causal correlation between TMP release and these biological outcomes. The same applies to the treatments that lead to perturbations in F-actin, which regulates a number of vesicular processes (including TCR recycling) during T cell activation. Hence the evidence, while suggesting a causal link between TCR shedding in association with TMPs and metabolic reprogramming and cell proliferation, does not formally prove it.

Point 4. The authors state that "in contrast to sAb treatment, T cells treated with iAb displayed a rapid recovery of surface TCRbeta (Fig.4C)". Substantial recovery can actually also be observed in sAb-activated cells. Also, if TCRs are mainly released in iAb-activated cells, recovery should involve de novo synthesis. Expression of the TCR-CD3 complex components should be tested by qRT-PCR.

Point 5. In introducing the T cell microvilli as a third mechanism of TCR release in addition of exosomes and microvesicles (lines 125-131) the authors do not consider synaptic TCR+ ectosomes, the release of which is also adhesion-dependent and requires TSG101.

Point 6. Despite the color code and the numbers, the legends of the histograms presented in figure 5 are not clear.

Point 7. Figure S1C. The Cyan arrowheads are not visible in the image.

Reviewer #3 (expertise in T cell activation, TCR exocytosis, immune synapse):

This is an interesting paper looking at the role of T cell microvillar/flipodial detachment in T cell activation. A particularly striking dataset relates to the effects of soluble anti-CD3 antibodies combined with adhesive substrates, fixed APC or F-actin depolymerisation in full T cell activation (proliferation and metabolic reprogramming). My only concern about these experiments is that in all cases the sAb is potentially adsorbed to the surfaces involved. Can the authors demonstrate that stimulatory sAb are not adsorbed to the surfaces involved, PLL, ICAM-1 coated plastic, fixed dendritic cells, etc. This could be done by treating the surfaces as in the assay conditions, washing out all the sAb and determining if the wells still have stimulatory activity based on antibody adsorption or binding to FcR, etc. If these controls work out then I would be convinced that the soluble antibodies combined with the various adhesive systems synergize to detach the projections and favour stimulation.

In the situation where latrunculin or cytochalasin facilitates activation by sAb, how does this support a role of detachment of projections as the projections are dependent upon F-actin for generation? Other studies suggest that such effects may be based on enhanced clustering of receptors following breakdown of cortical F-actin corrals. The new interpretation is different, but how does it support the trogocytic molting model?

RESPONSE TO REVIEWERS' COMMENTS

Response to Reviewer #1: (expertise in trogocytosis, extracellular vesicles):

Dear Reviewer,

We appreciate your insightful and critical comments regarding our manuscript entitled, “**Trogocytic-molting of T-cell microvilli controls T-cell clonal expansion (NCOMMS-22-23216-T)**”. In response to your comments, my colleagues and I have made significant changes to the manuscript, and we have addressed your comments, point by point. Our responses and accompanying revisions are as follows:

(General comments)

In this study, Park and co-workers address the role of T-cell microvilli shedding in T cell activation and metabolic reprogramming in response to TCR/CD3 signals. The study addresses a decades-long observation, which is that immobilized anti-CD3+anti-CD28 is more efficient than soluble anti-CD3+anti-CD28 to promote T cell activation. The authors had described TCR/CD3 complex at microvilli, and here they claim that microvilli shedding drives TCR upregulation and metabolic reprogramming towards clonal expansion.

Although of indisputable interest in the field and beyond, the study requires extensive experimentation and additional controls. **As the data stand, this is a nice, but phenomenological story, which may or may not be of general interest due to the fact that non-specific adhesion triggers the microvilli removal effect.** While this is important (and probably an unaddressed truth in the field), for the story to be truly relevant, one would expect this to be somewhat specific. In addition, the specific mechanisms that drive microvilli removal are cursorily examined, which is not acceptable at this stage.

Authors stimulate cells for hours or days and discuss their results as if these are physiological situations. In their system, there are no stop signals for interaction or adhesion: the use of immobilized antibodies cannot be assimilated to antigen-presenting cells. Of course, in the restricted volume of a cell culture well, cells are always subjected to the same conditions, which is not the case for circulating T cells.

Answer → We appreciate your overall critical comments on this paper. In this study, as you know, we have addressed two important issues of the long-held paradigm in T-cell biology.

First, immunologists have believed that TCR internalization is a major mechanism of surface TCR disappearance after T-cell activation. Therefore, many papers have dealt with studies on the internalization process of TCR and TCR complex. However, **our evidence demonstrates that, in reality, large amounts of TCRs and membrane proteins are released in the form of T-cell microvilli particles (TMPs) or synaptic ectosomes.** Indeed, sufficient data sets support that the major cause of surface TCR loss is due to TCR⁺ TMP (or TCR⁺ T-cell immunological synaptosomes; TISs) release, not TCR internalization.

As you noted in the first version of this manuscript, we only stimulated T cells for 3 h (either sAb or iAb), then washed and incubated with fresh medium for 3 – 5 days. The reason that we chose 3 h was because of the concept of providing sufficient time for stimulation. In this revised version, however, when the actual time for TCR⁺ TMP release was determined, we found that TMP release was much faster than expected and reached maximum at 5 to 10 min after stimulation. Since we pre-stained the membrane (PMS) or TCR β and measured PMS⁺ or TCR β ⁺ positive particles, it is unlikely that

exosomes are released within these time periods. Moreover, stimulation of T cells with soluble anti-CD3/28 resulted in no particle release, suggesting that exosomes are not involved in TCR downregulation on the cell surface. Therefore, in the revised manuscript, we chose T cell stimulation only for 5 to 30 min, and then further incubated for 3 – 5 days with fresh medium. In this paper, we also argue that the common notion that plate-immobilized anti-CD3/28 antibodies have stronger and longer-lasting signals than soluble anti-CD3/28 antibodies is a misguided paradigm. All our data show that the two stimuli are not intrinsically different. The only difference between the two conditions we found was whether there are microvilli shedding or not.

Second, TCR⁺ TMP (TIS) release has an important impact on T cell proliferation. Although you concerned about the specific mechanism for shedding process, we found that T cells can generate TMPs even when not in a specific adhesion process. Since T cells have already acquired specificity from the pMHC-TCR signal, we believe that specificity will not be needed again in the molting process. If the pMHC-TCR signal is correct, molting process occurs naturally by the adhesion process, which is thought to play a decisive role in the proliferation of T cells. Interestingly, as you see in Fig. 7d, LFA-1-KO T cells do not proliferate well on sAb (5 min stimulation and washing) + ICAM-1 because they cannot be adhered on ICAM-1. However, sAb-stimulated LFA-1-KO T cells placed on microvilli-specific anti-CD62L or non-specific on PLL (10 \times) significantly increase the release of TCR⁺TISs and enter cell cycle for proliferation comparable to wild-type T cells. These results strongly suggest that microvilli shedding is critical for T cell proliferation after TCR triggering.

In your comments, you suggested to examine the specific mechanisms that drive microvilli removal in revised manuscript. However, in fact, we aggressively studied the mechanism of microvilli release in T cells as the form of TMPs in a previous paper (Kim et al., Nature Com, 2018, 9:3630, 2018). From this point of view, we excluded the previous research contents as much as possible to avoid duplication of papers.

In addition, in order to develop a better storyline than in the first submission, we removed the previous Fig. 2a and b, moved previous Fig. 2c – e to Fig. 3a – c, moved previous Fig. 3a and b to Fig. 2a and b, and previous Fig. 3c and d to Fig. 3d and e in revised version.

The previous Fig. 3e was moved to Fig. S8 for better storyline.

A new result of TCR β ⁺ particle (TISs) release was added in Fig. 4b. To clarify the effects of sAb, we replaced the data of **previous 3 h stimulation to the data of 30 min stimulation** in Fig. 4c – g.

In Fig. 5a – e, we replaced the old figures (3 h stimulation) to new figures with 5 min stimulation of T cells with sAb. In addition, Fig. 6a and b are new figures that represent the relationship between molting and T-cell proliferation. Thus, previous Fig. 6a – d was moved to Fig. 8a – d.

Fig. 7b – d and Fig. 8e and f are new figures to improve our data quality and storyline. Figs. 7 and 8 in the first version was moved to Figs. 9 and 10.

In a revised version, we argue more strongly that the process of shedding on the T cell surface is an essential process for entering the cell cycle for proliferation, eventually clonal expansion.

MAJOR POINTS

Comment 1) Perhaps the most striking piece of data is that adhesion, even non-specific, increases T cell activation (measured as CD3zeta-GFP released particles). Is this dependent on adhesive strength? Does it vary depending on the type of adhesive receptor engaged? This is an important issue as, in the lymph node, T cells are constantly surrounded by non-specific, but potentially activating, adhesive

signals.

Answer and revision → (1) As we mentioned in the main text of manuscript, fragmentation of microvilli, which is eventually a form of TMP, requires both antigen-specific TCR signaling and adhesion-dependent trogocytic signals. As you see on Fig. 6a and b, without a specific TCR signaling, non-specific adhesion (ICAM-1, fibronectin, and VCAM-1) does not induce T cell activation for proliferation. However, TCR signaling in combination with proper adhesion strength can induce molting process that is critical for T cell proliferation (Fig. 4, 5, 6, 7, and 8).

(2) Although it was commonly accepted that plate-bound, immobilized anti-CD3/28 is believed to work better as it mimics cell-cell contact and may also guarantee the chance for the cells to interact with immobilized anti-CD3/28 antibodies on the surface of the plate, we found that the two conditions are not essentially different in terms of TCR signaling intensity, at least within 30 min (see Fig. 4a). To our surprise, even a **brief stimulation (5 min of sAb)** was sufficient to initiate metabolic reprogramming for proliferation when paralleled with the molting process (Fig. 4, 5, and 7).

(3) In a revised version, a new attempt was made to obtain *in vivo* evidence of TCR release upon cognate interaction with antigen-presenting cells (APCs). To this end, wild-type C57BL/6 mice pre-immunized for 48 h with either OVA257-264 (MHC class I) or OVA323-339 (MHC class II) peptides were injected with OTII TCR T cells stained with CTV and anti-TCR β -Fab-Alexa 488. Significant downregulation of TCR β^+ intensity was only observed in OTII TCR T cells from OVA323-339- but not OVA257-264-immunized mice, demonstrating that membrane TCRs are released into the cognate DCs *in vivo* in a specific manner (see Fig. 1e).

Comment 2) Imaging studies are usually descriptive, with **only one cell analyzed**; and stimulation for **3 h cannot be considered as a short period** to study changes in the membrane and cytoskeleton of T cells. Why is the TCRbeta staining in CD4 T cells from OVA+anti-LFA-1 Ab condition similar to the OVA w/o anti-LFA-1 Ab condition (but different from no peptide contact with dendritic cells; Fig 1E)? The dendritic cell does not seem to internalize any particle received by the T cell.

Answer and revision → (1) As you find in Fig. 1a, surface TCR loss was mostly occurred within 30 min of stimulation. In our new version, moreover, we found that TCR $^+$ or PMS $^+$ TMPs are maximally released within 5 – 10 min of stimulation on iAb. Therefore, we replaced some previous data and added new data with the 5 – 30 min stimulation method. See, Figs 4 – 7.

(2) As you know, a single cell image was presented as a representative image, but at least 50 cells were analyzed for each experiments. Additionally, some data are from a flow cytometer that has analyzed thousands of cells or more. However, according to your comment, we quantitated the images of more than 50 cells as shown in Fig. 2a, Fig. S1, S3, and S4.

(3) We presented live images showing the release of TCR ζ -GFP $^+$ particles during OTII T-cell interaction with antigen (OVA)-bearing dendritic cells (DCs) in Fig. S3b. In fact, since the previous Fig. 1e was almost similar to Fig. S3b except that the anti-TCR β Fab was used, we removed the previous Fig. 1e to minimize the redundancy of the results. Instead, we have added new results of *in vivo* evidence of TCR downregulation via TCR $^+$ TMP release in Fig. 1e. Please see Fig. 1e, and Fig. S3a and b.

Comment 3) Does a surface coated with ICAM-1 alone promote TCR shedding? What about VCAM-1 or fibronectin?

Answer and revision → Thank you for your excellent comment. As you see in Fig. 6a and b, T-cell adhesion on various adhesion molecules by itself is not sufficient to promote TCR shedding while some extents are released. As we consistently assert, TCR shedding necessarily requires TCR activation. As you seen in Fig. 6a and b, T cell proliferation is evidently correlated with the release of TCRβ⁺ particles.

Comment 4) A major issue with this paper is that the mechanistic connection between cell adhesion, TCR/CD3 triggering and microvilli release is barely worked out. The only mechanistic insight into this crucial issue in the story is provided in Fig. 6A. What is triggering the engagement of the abscission (cutting) complex (containing Vps4)? Is it adhesion? Is it the TCR? **For this part to make a real contribution, this pathway needs to be deciphered in more detail.**

Answer and revision → In our previous report, we aggressively studied the mechanism of microvilli release in T cells as the form of TMPs (Kim et al., Nature Com, 2018, 9:3630, 2018). From this point of view, we excluded the previous research contents as much as possible to avoid duplication of papers. In a previous work, we found that arrestin-domain-containing protein-1 (ARRDC1), TSG101, and Vps4a/b are highly enriched in TMPs. Interestingly, we found that overexpression of ARRDC1 recruits TSG101 from the cytosol to the membrane and this protein moved and released similarly to TCRβ during T cell synapse and kinapse (left figures a – c). Knockdown of ARRDC1 or Vps4 significantly reduced TMP release. However, to minimize the redundancy of data, we referred to our previous paper and did not perform the same experiments again.

Instead, as you see in Fig. 7d, we performed new experiments utilizing LFA-1-KO T cells and verified that adhesion-dependent TIS release is essential for T cell proliferation after TCR triggering. LFA-1-KO T cells do not proliferate well on sAb (5 min stimulation and washing) + ICAM-1 because they are not adhering on ICAM-1. However, specific adhesion on

microvilli-specific anti-CD62L can increase TCR⁺TIS release and molted T cells enter the cell cycle for proliferation comparable to wild-type T cells. See Fig. 7d.

Comment 5) Authors analyze naïve CD3 T cells, and therefore CD4 and CD8 cells indistinctly. These cells show very different behavior when referring to TCR activation, adhesion and mechanotransduction. In Fig 3D, **the CD4 and CD8 receptors are barely lost upon activation with pOVA and ICAM-1**; does this mean that they are not binding correctly to the MHC? Are the microvilli devoid of CD4 or CD8?

Answer and revision → Thank you for your excellent comment. Although we used mixed population of CD4 and CD8 T cells to perform the experiments, we also examined each cell population after sorting CD4 and CD8 T cells separately, as shown in Fig. S10. The reason that the membrane CD4 and CD8 are not reduced as much as TCRβ is that not all CD4 and CD8 molecules are likely co-localized with TCRβ in microvilli. For your reference, we show the confocal results of TCR (green)- and CD4 (red)-pre-stained T cells stimulated with iAb or sAb (left panel). See new Fig. S6.

As shown in the left panel (Fig. S6), most TCRs are released (iAb) or internalized (sAb) after T-cell activation. In contrast, some CD4 molecules still remain on the cell membrane under both conditions. These results may suggest that some of the CD4 molecules are not localized at the microvilli tips.

Interestingly, a previous report of Mark M. Davis group demonstrated that the co-receptor CD4 is expressed in distinct nanoclusters and does not co-localize with TCR and active protein tyrosine kinase p56lck (Roh et al., PNAS, 112 (13) E1604-E1613). In contrast, recent report

of Gilad Haran group demonstrated that CD4 and CD8 molecules are exclusively enriched in microvilli on T cells (Ghosh et al., Cell Rep. 2020 Mar 10;30(10):3434-3447). Based on our works, it seems that not all co-receptors are localized at the microvilli tips. To minimize any confusion, we only revealed the result of CD4 T cells in Fig. 3e.

Comment 6) That LatA treatment impairs T cell activation is not novel, but the reasoning the authors use to justify this result is somewhat skewed. At least with CK636, the authors can rule out the involvement of dendritic actin polymerization in the process, but **the Jask and LatA experiments do not make a compelling enough argument to sustain that microvilli are critical for T cell activation by IS formation**, but once TCR has been engaged, shedding is essential.

Answer and revision → After T cell activation, global actin rearrangement is essential. Thus, disruption of actin network at this stage results in weak TCR signaling. As expected, since LatA and CytD are major actin-disrupting agents, treatment of T cells with these reagents significantly attenuated T cell activation **on iAb** (See Fig. 8g and S12). In fact, these results are very consistent with many reports in the literature. However, we found that after TCR activation by sAb (for 5 min), LatA dramatically induced TCR⁺TIS release more than LatA alone (Fig. 7a and b). **This may be due to the fact that TCR activation tends to synergize LatA-induced microvilli release. This result is consistent with the previous reports demonstrating that actin-disrupting agents can rapidly release the coated vesicles from the plasma membrane (Exp Cell Res, 1981, 135:431). In addition, this result suggests that TCR signaling evoked by sAb (for 5 min) is sufficient to induce T cells to enter the cell cycle for proliferation.** On the contrary, JPK clearly reduced TCR⁺TIS release in activated T cells, as determined by flow cytometer analysis (Fig. 8e).

Because you pointed out somehow side-effect of these actin reagents, we used LFA-1 KO mice. T cells from LFA-1 KO mice contain intact surface microvilli but do not bind to ICAM-1. In this

situation, these T cells do not produce TMPs and do not proliferate well. However, these T cells can bind non-specifically to PLL (10×) or specifically to anti-CD62L (L-selectin). These two conditions can induce TMP release after T cell activation. Stimulation of T cells with sAb for only 5 min was sufficient for proliferation when the stimulated T cells were placed on non-specifically PLL (10×) or on microvilli-specific anti-CD62L (L-selectin) (Fig. 5a – c and Fig. 7d).

Comment 7) In fact, the data reported here could be **used to argue that TCR downregulation is essential for T cell activation**. What happens is the authors target TCR expression transiently with, for example, siRNA, after activation? Do cells undergo a proliferative repression followed by increased activation? Probably, microvilli in cells and TMPs recovered from surface are dissimilar in components.

Answer and revision → If you see Fig. 2a and Fig. 4b, there are dramatic difference between sAb and iAb in terms of TCR internalization and TCR⁺ TIS release. Please consider that the intensity of TCR signals evoked by two conditions (sAb and iAb) is not different (see Fig. 4a). Thus, the only difference between the two conditions is whether the T cells are adhered on the matrices (iAb, pMHC+IC1, pMHC+VC1, sAb+PLL, sAb+IC1, sAb+anti-CD62L or ConA+ all adhesion matrices) or otherwise suspended (sAb). In the suspended condition, TCR activation does not secrete TISs, but rather the TCRs are internalized.

Since intact TCR signaling (induced by antibodies or other stimulations including pMHC and ConA) is important to trigger TCR⁺ TIS release in T cells, transient downregulation of TCR by siRNA may result in insufficient T cell activation states such as anergy states. This condition may reduce molting process. We think that this situation is similar to the current experiment in which LatA was pretreated prior to activation in pMHC+ICAM-1 (Fig. 8e and Fig. S12). In addition, the released TCR⁺ TISs are not just TCRs, but a large number of T cell proteins and lipids are simultaneously released. In this case, it seems meaningless to down-regulate only a single protein, i.e., TCR, rather, it is necessary to control the expression of numerous TIS proteins, which is practically impossible. We hope you accept our answers.

Comment 8) In general, if microvilli are sensory “organs” that cells use during IS formation and are shed for proper T cell activation, **why do activated cells display many more microvilli after 72h?** (Fig. 4B). In principle, those cells would be already activated and would have little need for additional microvilli. In the same vein, if those cells are re-stimulated, do they shed those microvilli as well? Or those “**2nd generation**” microvilli are somewhat more resistant to shedding?

Answer and revision → Thank you for your interesting question. As you know, one of the important features of adaptive immunity is the rapid and strong defense of our body against infection, and T cells are important components of adaptive immunity. After clonal expansion, T cells differentiate into effector T cells with more microvilli on their surface. Effector cells must rapidly recognize antigens on APCs or target cells. Therefore, more microvilli (TCRs) are essentially required.

As you see in the left figure (Fig. S9), the effector T cells released more TCR⁺TISs than naïve T cells. This is very likely because effector T cells have more microvilli on their surface. However, TIS release at this stage may not be linked with T cell proliferation as they are already differentiated into effector T cells. Instead, T cells at this stage can become more potently activated and thus may eventually die by activation-induced cell death (AICD). Notably, an increased number of

microvilli and increased TCR expression are markers for antigen-experienced or memory T cells. Further studies are underway to explore whether membrane molting or TCR internalization is coupled with the conversion of effector T cells to a memory phenotype.

Comment 9) Why would TMPs contain metabolic components? Or this is just a representation of the metabolic reprogramming of activated cells? If so, shouldn't these changes be observed in total cell lysates? The rationale for performing MS, GO and the like on TMP content seems weak.

Answer and revision → Since TMPs (TISs) are primarily derived from the plasma membrane, TMPs are composed of numerous membrane-derived proteins, cytokines, lipids, and other functional substances that potentially play on T cell functions as well as their survival. Therefore, it is not surprising that TMPs (TISs) contain many substances related to metabolism. However, since TMPs are shed from the T-cell membrane surface in a rapid response (~5 – 10 minutes), TMP constituents themselves are unlikely to reflect the reprogramming state of T cell metabolism. The metabolic processes would rather be initiated after TMPs are released in T cells. However, **the reason why we analyzed the components constituting TMPs is to understand at the molecular level what happens in T cells although these essential components are released from T cells.** In the process of molting, snakes or insects release their important outer layer substances, but rather, the organism grows. However, we do not think that shedding material represents metabolic reprogramming occurring in the body of snakes or insects.

Interestingly, when we carefully examined the data of LC-MS/MS, we found that ferritin heavy and light chain are extraordinary enriched in TMPs, suggesting that ferritin-iron complex are released via shedding process during T cell activation. It has been known that T cell activation leads to a drastic loss of intracellular iron (Fe²⁺) (Yarosz et al., J Immunol, 2020, 204:1708). The authors demonstrated that high levels of cellular labile iron pool (LIP) prevents T-cell proliferation, implying that T cells develop a distinct mechanism to prevent iron-overload during T-cell activation. Fe²⁺ are released from ferritin by degradation pathway via NCOA4 to maintain intracellular iron homeostasis (Zhang et al., Mol Neurobiol, 2021, https://doi.org/10.1007/s12035-020-02277-7). Based on our current works, we therefore think that ferritin and iron complex are rapidly secreted to the extracellular space through TISs, which can temporarily reduce the intracellular iron pools, possibly turning on T cell proliferation. Our results further suggest that TIS release by molting process is essential for the T cell

survival during activation. Although we do not add the ferritin results at this time, further studies are underway to understand the biological meaning of iron loss through TISs.

Comment 10) The correlation between the type of metabolic reprogramming (FAS or FAO) and microvilli shedding is, as shown, phenomenological. **The authors need to set an efficient intervention point that prevents microvilli shedding in adhered cells (not a cytoskeletal drug) and probe metabolic reprogramming in those cells.**

Answer and revision → In the main figures, we have already presented sufficient results that activated T-cell adhesion to the matrix induces TCR⁺TIS release, regardless of stimulation method (Fig. 5, 6, 7, and 8). In addition, the cytoskeletal drugs LatA and CytD evidently mimicked the situation of adhesion-induced TCR⁺TIS release and consequent T cell proliferation. Moreover, our results are consistent with the previous reports demonstrating that actin disrupting agents can rapidly release the coated vesicles from the plasma membrane (Exp Cell Res, 1981, 135:431 and Int J Nanomedicine, 2021, 16:8485). Thus, we believe that the use of a cytoskeletal drug latA is the right way to most clearly compare the difference between the sAb and iAb in terms of looking at global gene expression regulation.

As you see in Fig. S14, we also tried to compare the gene expression regulations mediated by Concanavalin A (ConA) vs ConA+PLL. We found that only ConA+PLL, which dramatically release TCR⁺ TISs, induced the upregulation of cell proliferation (enrichment test p values <10⁻⁸) (Figs. S14a and S14b) and cellular biomass synthesis (e.g., cholesterol, nucleotide, amino acids, and fatty acids) (Figs. S14c and S14d; fGSEA p values <0.05). In conclusion, all of the present results clearly demonstrate that T cell proliferation is correlated with the amount of released TCR⁺TISs. Thus, we do not think we can obtain new insight if we perform new experiments with different intervention point.

One possibility is to use LFA-1-KO T cells to interpret the results of global gene expression regulation when activated T cells are placed on different adhesion matrices including ICAM-1 and anti-CD62L Ab. Again, however, we do not think it would be much different from using LatA. I hope you understand this situation.

Minor comments:

Comment 11) Authors should carefully check the references. References 7 and 17 are duplicated; and reference 17 is mistakenly written.

Revision → Thank you. We changed it.

Comment 12) The titles in figure legends do not correspond to the experiments showed; e.g.: in figure 2, there is no APC involved.

Revision → In a revised manuscript, we polished many parts and added new results of experiments. Thank you.

Comment 13) The reduced resolution in Z for classical confocal microscopies does not allow proper analysis of co-localization or localization

Answer → For most confocal analysis, we used LSM 880 (Carl Zeiss, Germany) equipped with an

Airyscan with a resolution of 1.7-fold higher than that of a classical confocal microscope. We think this resolution is sufficient to analyze most of the work performed in this study.

Comment 14) Fig S8B. Yellow is not included in the figure.

Revision → We changed it.

Comment 15) The use of orange and blue color in the Fig. 5 for different parameters is misleading.

Revision → Thank you. Many results have been added or subtracted in Fig. 5.

Comment 16) Anti-CD62L is not an irrelevant antibody (line 234) if it has so much effect on cells as observed in the release of microvilli, OCR/ECAR experiments, and on the recovery of TCR and in cell division when combined with sAb (Fig. 5).

Revision → We polished it.

Response to Reviewer #2: (expertise in immune synapse, immune cell signaling):

Dear Reviewer,

We appreciate your insightful and critical comments regarding our manuscript entitled, “**Trogocytic-molting of T-cell microvilli controls T-cell clonal expansion (NCOMMS-22-23216-T)**”. In response to your comments, my colleagues and I have made significant changes to the manuscript, and we have addressed your comments, point by point. Our responses and accompanying revisions are as follows:

In this report the authors address the role of TCR shedding via T cell microvilli particles (TMP) in the decrease in surface TCR that occurs following TCR engagement. In the current view, TCR downregulation is the result of ligand-dependent endocytosis, which is followed by either degradation or recycling to the T cell surface. Here the authors show that, when T cells are activated by immobilized anti-CD3 mAb, or MHC-bound peptide plus ICAM-1 on supported lipid bilayers (SLB), or by cognate dendritic cells (DC), they downregulate surface TCRs in an adhesion-dependent manner by releasing TCR+ TMPs rather than by TCR internalization. They show that TCR shedding in association with TMPs leads to an enhancement in TCR expression, metabolic upregulation (associated with fatty acid synthesis), and enhanced cell proliferation and survival. Based on these findings, they propose that TCR release through microvilli shedding triggers an intrinsic circuitry that sets the conditions for clonal expansion.

This report sheds light into the mechanisms that regulate TCR levels at the cell surface and hence the ability of the T cell to respond to cognate ligand. The impressive amount of work presented in the manuscript, which involved a variety of approaches, is of excellent quality and overall supports the authors' conclusions. **However, the results do not fully answer one of the main questions, i.e. the role of TCR release versus endocytosis in TCR downregulation.** The authors have carried out all the experiments using two main activation settings (in some cases complemented by additional ones, e.g. SLBs or cognate DCs): "soluble" anti-CD3 mAb (sAb), that involves a subsequent step of cross-linking by secondary Abs, and immobilized (plate-bound) anti-CD3 mAb (iAb), in both cases in the presence of anti-CD28 mAb. The results clearly show that, starting from a different primary mechanism of TCR downregulation (endocytosis vs TMP shedding, respectively), these conditions lead to very different biological outcomes. But if, as the authors underscore, iAbs represent the condition closer to physiological stimulation compared to sAbs (as also supported by the results obtained using the SLB or DC activation settings), the data obtained using sAbs are a major confounding factor. To have clear information on the respective contribution of TCR endocytosis vs shedding in T cell activation, **the two pathways should be analyzed under the same condition, namely the more physiological iAb setting.** This has been done, but should be done more quantitatively. Additionally, the authors are challenging a model that has been established over decades with the contribution of many research groups. **Hence the role of TCR endocytosis, as well as recycling (which has not been considered), should be addressed more in depth, experimentally and in the discussion.**

General messages to the reviewer #2 → We appreciate your overall critical comments on this paper. As you mentioned, in the current view, TCR downregulation is the result of ligand-dependent endocytosis, which is followed by either degradation or recycling to the T cell surface. Therefore, we agree that our works are very challenging and not easy to overcome a model that has been established over decades with the contribution of many research groups. But my question at this point is, why haven't immunologists seriously considered so far how much amount of membrane fractions are released during T-cell contact with APCs? Indeed, an extensive body of evidence indicates that

surface proteins are commonly transferred between immune cells *in vitro* and *in vivo*. This cell-surface protein transfer has been referred to “absorption”, “nibbling”, or “trogocytosis” and discovered almost 40 years ago. Scientists have thought that trogocytosis is a phenomenon in which only a small portion of cell membrane fragments can be “torn off” by the tensile force generated when cells try to break apart. However, why was it not considered at the time that trogocytosis could be a mechanism of TCR downregulation? In our previous paper published in Nature com (Kim et al., 2018, 9:3630), we recognized that a lot of membrane microvilli particles containing TCR molecules are released during T cell interaction with APCs, pMHC+IC1, or immobilized anti-CD3/28. The results of our previous works motivated us to investigate whether TCR⁺ TMP release is related with TCR downregulation.

Although you said that the data obtained using sAbs are a major confounding factor, we could obtain many interesting results from sAb condition. As you know, sAb is not a physiological condition. However, we recognized that both iAb and sAb are essentially not different in terms of the intensity of TCR signaling (Fig. 4a). The only difference between the two conditions was whether microvilli shedding was present (Fig. 2a and Fig. 4b). In the first version of this manuscript, unless otherwise indicated, we chose 3 h stimulation in both conditions (sAb or iAb), followed by washing, and incubation for 3 – 5 days with fresh media. The reason that 3 h was chosen was due to the concept of providing sufficient stimulation. In the revised manuscript, however, when we examined the actual release time of TCR⁺ TMPs, we found that the TMP release was faster than expected and was almost maximal between 5 to 10 min of stimulation. Since we pre-stained the membrane (PMS) or TCR β , it is unlikely that exosomes are released within these time periods. Moreover, stimulation of T cells with sAbs produced few particles, suggesting that exosomes are less likely to be involved in cell-surface TCR downregulation. Therefore, in the revised manuscript, we chose T cell stimulation only for 5 to 30 min, and then further incubated for 3 – 5 days with fresh medium. In this paper, we also argue that the common notion that plate-immobilized anti-CD3/28 antibodies have stronger and longer-lasting signals than soluble anti-CD3/28 antibodies is a misguided paradigm.

As shown in Fig. S1, all T cells released TCR⁺TISs to the bottom of the culture plate with or without endocytosis inhibitors and this was consequently related to a decrease of surface TCR fluorescence intensity. Once TISs are released, activated T cells may have a license to enter the cell cycle for proliferation. As shown in Fig. 6, in more physiological conditions, the amount of shedding (or TCR⁺ particle release) is exactly correlated with the TCR disappearance on the membrane and T cell proliferation. Interestingly, as you see in Fig. 7d, LFA-1-KO T cells do not proliferate well on sAb (5 min)/ICAM-1 because they are not adhered on ICAM-1 and thus membrane shedding does not occur. However, non-specific adhesion of LFA-1-KO T cells on PLL (10 \times) or adhesion on anti-CD62L Ab significantly induced TCR⁺ particles, which allowed proliferation comparable to wild-type T cells.

In addition, in order to develop a better storyline than in the first submission, we removed the previous Fig. 2a and b, moved previous Fig. 2c – e to Fig. 3a – c, moved previous Fig. 3a and b to Fig. 2a and b, and previous Fig. 3c and d to Fig. 3d and e in revised version.

The previous Fig. 3e was moved to Fig. S8 for better storyline.

A new result of TCR β ⁺ particle (TISs) release was added in Fig. 4b. To clarify the effects of sAb, we replaced the data of **previous 3 h stimulation to the data of 30 min stimulation** in Fig. 4c – g.

In Fig. 5a – e, we replaced the old figures (3 h stimulation) to new figures with 5 min stimulation of T cells with sAb. In addition, Fig. 6a and b are new figures that represent the relationship between molting and T-cell proliferation. Thus, previous Fig. 6a – d was moved to Fig. 8a – d. The previous Fig. 6e was moved to Fig. S12.

Fig. 7b – d and Fig. 8e and f are new figures to improve our data quality and storyline. Figs. 7 and 8 in

the first version was moved to Figs. 9 and 10.

In a revised version, we argue more strongly than before that the process of shedding on the T cell surface is an essential process for entering the cell cycle for proliferation, eventually clonal expansion.

Specific points

Comment 1) Figure 2B. Based on size of the particles detected by quantitative flow cytometry in the culture supernatants of iAb-stimulated cells the authors propose that the 100 nm-range population corresponds to TCR+ TMPs. This should be confirmed by staining with anti-TCR antibodies as well as antibodies to microvilli markers (e.g. Vstm5)

Answer and revision → We appreciate your excellent comment. Indeed, measurement of small vesicles by flow cytometer is more convenient and accurate ways than the data analysis of confocal images. Although we usually calculated PMS⁺ particles as this fluorophore can stain membrane, we alternatively used anti-TCRβ Fab-Alexa488 to directly measure TCR⁺ particles. For a better story development than before, we moved what was originally in Fig. 2b to Fig. 4b. In addition, we also replaced previous figures with new figures. See Figs 4 – 8. Since Vstm5_GFP is only used for T blasts, we could not use this vector for naïve T cell experiments.

Comment 2) Figure S1. In panel A the authors used inhibitors of clathrin-dependent endocytosis as well as proteasome and lysosome-dependent degradation to address the role of these pathways, previously implicated in TCR downregulation, in the decrease in surface TCR observed in iAb-stimulated cells. Based on the lack of effect of these inhibitors, the authors conclude that TCRs are mainly released during T cell adhesion. **Have the authors checked that these treatments affect the (smaller) decrease in surface TCR in sAb-activated cells, which based on their data should mainly be the endocytic one? Also, TCR internalization has been reported to be regulated in part by a clathrin-independent pathway that leads to their incorporation into an endocytic network marked by flotillins.** This network is essential for TCR recycling. This pathway should also be ruled out in iAb-

activated cells.

Answer and revision → Thank you for your suggestion. Based on our experimental setting, as you see in Fig. S1a, various inhibitors had no significant effect in the change in intensity of pre-stained TCRβ-fluorescence in the sAb condition. Notably, the confocal data indicate that some extents of TCRβ-fluorescence are not endocytosed in the presence of endocytosis inhibitors (CPZ and PP2) but are retained in the membrane under sAb condition (Fig. S1b). However, dramatic release of TCR⁺ particles (green color) is seen under iAb condition, regardless of the presence of inhibitors. These results suggest that the process of T-cell membrane shedding is independent of the mechanism of receptor endocytosis.

We also applied flotillin 1 (flot1) siRNA to test whether flotillin-pathway is involved in the molting process of TCR⁺ microvilli. As shown in the figure on the left (See Fig. S2), knockdown of flot1 did not affect the TCR shedding process and pre-stained TCRβ-fluorescent intensity induced by iAb or sAb.

Comment 3) Figure 6C. The authors show that Vps4a/b knockdown leads to attenuated metabolic reprogramming and reduced T cell proliferation. How stable is the knockdown over time? Also, Vps4, together with the ESCRT complex, is not only essential for microvilli fission but also for the generation of other membrane vesicles, including **multivesicular body-derived extracellular vesicles and synaptic ectosomes**. Hence these data do not provide a final causal correlation between TMP release and these biological outcomes. The same applies to the treatments that lead to perturbations in F-actin, which regulates a number of vesicular processes (including TCR recycling) during T cell activation. Hence the evidence, while suggesting a causal link between TCR shedding in association with TMPs and metabolic reprogramming and cell proliferation, does not formally prove it.

Answer and revision → We appreciate your important comment. I totally agree with your opinion. Most of approaches we applied was also somewhat overlapped with the release of other membrane vesicles. It was not easy to select systems or reagents which are selective only for microvilli shedding. In a revised version, to derive more direct connection between the microvilli shedding (TMP release) and T-cell division, we utilized LFA-1-KO T cells (Fig. 7d). These cells do not bind well on sAb (5 min)+ICAM-1, but significantly bind to sAb+plate-coated anti-CD62L, as these cells have intact microvilli and CD62L. In addition, these T cells also can adhere non-specifically on PLL (10×). These two conditions generate higher amount of TMPs and the release of TMPs is correlated with the proliferation of LFA-1-KO T cells (Fig. 7d). These results demonstrate that microvilli shedding is critical for entering cell cycle for proliferation in activated T cells. Secondary, as you suggested, we alternatively used anti-TCRβ Fab-Alexa488 to directly measure TCR⁺ particles. This also could be a direct evidence of microvilli shedding. If T cells are treated with JPK, it significantly reduces TCRβ⁺ particles (Fig. 8e). This suggests that JPK inhibits the release of membrane particles, but not intracellular vesicles. Moreover, we could not find any vesicles from sAb-activated T cells. Third, our results (increased PMS⁺ particle release by LatA or CytD) are consistent with the previous reports demonstrating that actin disrupting agents can rapidly release the coated vesicles from the plasma membrane (Exp Cell Res, 1981, 135:431 and Int J Nanomedicine, 2021, 16:8485).

Comment 4) The authors state that "in contrast to sAb treatment, T cells treated with iAb displayed a rapid recovery of surface TCRbeta (Fig.4C)". Substantial recovery can actually also be observed in sAb-activated cells. **Also, if TCRs are mainly released in iAb-activated cells, recovery should involve de novo synthesis. Expression of the TCR-CD3 complex components should be tested by qRT-PCR.**

Answer and revision → Thank you for your suggestion. We think that the reason that some T cells treated with sAb also restore surface TCR is due to receptor recycling instead of *de novo* synthesis of the TCR gene. We performed qRT-PCR according to your suggestion. T cells placed on iAb significantly upregulated *de novo* synthesis of TCR gene 1 h after stimulation, which correlated with the increased metabolic reprogramming (see Fig. 4h – j). In contrast, no significant upregulation of TCR gene expression was observed by sAb treatment (Fig. 4j).

Comment 5) In introducing the T cell microvilli as a third mechanism of TCR release in addition of exosomes and microvesicles (lines 125-131) the authors do not consider synaptic TCR+ ectosomes, the release of which is also adhesion-dependent and requires TSG101.

Answer and revision → Despite the possibility of different types of microvesicles being released from activated T cells, we suggest an overlapping origin of TCR-enriched microvesicles and synaptic ectosomes with T cell-derived microvilli particles (TMPs). We therefore called the membrane particles released from the activated T cells on adhesion substrates as T cell immunological synaptosomes (TISs). We added this sentence in the Result section. Lines 153 – 156. In addition, in the body of text, we changed the word “TMPs” to “TISs”.

Comment 6) Despite the color code and the numbers, the legends of the histograms presented in figure 5 are not clear.

Answer and revision → We are sorry for the confusion. In the revised version, many parts of Figure 5 were replaced with new figures. Thank you.

Comment 7) Figure S1C. The Cyan arrowheads are not visible in the image.

Answer and revision → Fig. S1b and c are moved to the Fig. S3a and b. You will be able to easily recognize the cyan arrowheads.

Response to Reviewer #3: (expertise in T cell activation, TCR exocytosis, immune synapse):

Dear Reviewer,

We appreciate your insightful and critical comments regarding our manuscript entitled, “**Trogocytic-molting of T-cell microvilli controls T-cell clonal expansion (NCOMMS-22-23216-T)**”. In response to your comments, my colleagues and I have made significant changes to the manuscript, and we have addressed your comments, point by point. Our responses and accompanying revisions are as follows:

Comment 1) This is an interesting paper looking at the role of T cell microvillar/flipodial detachment in T cell activation. A particularly striking dataset relates to the effects of soluble anti-CD3 antibodies combined with adhesive substrates, fixed APC or F-actin depolymerisation in full T cell activation (proliferation and metabolic reprogramming). My only concern about these experiments is that in all cases the sAb is potentially adsorbed to the surfaces involved. Can the authors demonstrate that stimulatory sAbs are not adsorbed to the surfaces involved, PLL, ICAM-1 coated plastic, fixed dendritic cells, etc. This could be done by treating the surfaces as in the assay conditions, washing out all the sAb and determining if the wells still have stimulatory activity based on antibody adsorption or binding to FcR, etc. If these controls work out then I would be convinced that the soluble antibodies combined with the various adhesive systems synergize to detach the projections and favor stimulation.

Answer and revision → Thank you for your nice comments. According to your comment, we performed the experiment on whether sAb is potentially adsorbed to the surfaces, affecting our experimental results. Based on ELISA assay, we observed that treated sAb had no significant effects on the experimental conditions (data not shown).

Meanwhile, during the revision experiments, we found that the two conditions (iAb and sAb) were intrinsically not different in terms of the intensity of TCR signaling, at least within 30 min (Fig. 4a). Very interestingly, even a brief stimulation (5 min) of T cells with sAb was sufficient to induce molting process if the cells were placed on adhesion matrices (PLL10 \times , ICAM-1, anti-CD62L) or incubated with LatA (Figs. 5 and 7).

Although we performed the experiments you mentioned, we cannot upload the results to this rebuttal letter because, due to the policy of nature publishing group, all data in the rebuttal letter must be presented in the main Figures. However, we do not believe this figure is necessary for the storyline, as we found that only 5 min stimulation with sAb is sufficient to activate T cells for proliferation. Hope you understand our answer.

Comment 2) In the situation where latrunculin or cytochalasin facilitates activation by sAb, how does this support a role of detachment of projections as the projections are dependent upon F-actin for generation? Other studies suggest that such effects may be based on enhanced clustering of receptors following breakdown of cortical F-actin corrals. The new interpretation is different, but how does it support the trogocytic molting model?

Answer and revision → First of all, LatA or CytD can induce shedding of microvilli as observed by SEM (Fig. 7a). In addition, they could significantly increase the release of PMS⁺ or TCR⁺ particles from sAb-stimulated T cells, although the exact reason is unclear. As shown in Fig. 4b, since activated T cells rarely produce other types of vesicles except for PMS⁺ or TCR⁺ particles in a short period of time, we reasonably infer that these two actin-disrupting agents act more specifically on microvilli shedding. Indeed, our results are consistent with the previous reports demonstrating that actin

disrupting agents can rapidly release the coated vesicles from the plasma membrane (Exp Cell Res, 1981, 135:431 and Int J Nanomedicine, 2021, 16:8485). Furthermore, since we stained the T cells with PMS or anti-TCR β -Fab-Alexa488, PMS⁺ or TCR⁺ fluorescent intensities suggest that parts of T-cell membrane (microvilli) are released.

A previous paper which we know is that actin disrupting agent CytD inhibits clustering of TCRs rather than enhancing them (MCB, 2004, 24, p1628–1639). In the paper, the authors demonstrated that CytD prolonged elevation of intracellular Ca²⁺, leading to persistent NFAT nuclear duration. We think the elevation of intracellular Ca² levels and persistent NFAT activity are also associated with increased release of TCR⁺ TISs from T cells by actin-disrupting agents.

In a revised manuscript, we have changed a lot of parts and some experiments were redesigned. Thus, the present results are novel and make significant conceptual advances in the field of T cell immunology, especially the mechanism of clonal expansion. Thank you.

REVIEWER COMMENTS

Reviewer #1 (expert in trogocytosis, extracellular vesicles):

I commend the authors for the work they performed to revise the manuscript. The new experimentation has very much improved its scientific content and better support their conclusions. In particular, the experiments including LFA-1-deficient cells are well-designed and provided clear results. Also, the use of shorter time points and different substrates and adhesion molecules support the authors' statements.

I only have a few remaining issues that need to be addressed:

1. The in vivo experiments demonstrate that TCRs are released from T cells, but uptake by DCs in these conditions is not addressed. These experiments do not need to be carried out for this already long story, but the conclusions regarding this part (lines 132-133) need to be toned down.
2. It is surprising that the authors observe no cytoplasmic presence of TCRzeta-GFP (Lines 140-142). This is somewhat different from what is seen with the endogenous protein. Please, specify and comment.
3. Please, quantify the results obtained by Western blot (densitometry and point-by-point representation including the number of replicates) throughout the manuscript.
4. The OCR/ECAR ratio does not add any new information, it is not mentioned by authors and can be misleading to the readers. Authors should remove it (Fig. 4). Are there differences in OCR and ECAR between NT, sAb and iAb? Please, include stats in the figure.
5. Are the cells activated with iAb consuming less lipids than sAb? Authors should add an OCR/ECAR dataset using lipids as energy fuel to confirm this important issue.

Reviewer #2 (expert in immune synapse, immune cell signaling):

The authors addressed satisfactorily all the issues raised in my previous review with substantial experimental work.

I have a minor point that I noticed only on reading the revised manuscript to find the revised parts matching my critiques. Commenting on figure 4a, the authors state that there are no differences in signaling elicited by sAb versus iAb. This is not totally true in this figure (but it is clear in figure S8, which however does not include in the legend the sAb/iAb concentration used or the time of stimulation). For example, the phosphorylation of PKC θ appears more robust and sustained in response to sAb (10 microgram/ml condition). At the same Ab concentration pErk and pPLC γ appear lower in cells stimulated with sAb, but this might not be the case if the signals are normalized to the actin loading control, which is not homogeneous across the samples and is lowest at the 30 min time point of the 10 microgram/ml sAb sample. To clarify this point the immunoblots should be quantified over multiple experiments and quantifications shown (with stats).

Reviewer #3 (T cell activation, TCR exocytosis, immune synapse):

The authors have responded adequately to my limited comments from the earlier review. They have performed ELISA type assays to look for deposition of sAb to well surfaces, leading to low density of iAb. I would have preferred if they use T cells to detect functional levels of iAb follow sAb based assays because T cells are more sensitive, but the Elisa is a standard approach. I was not aware that latrunculin A leads to shedding of microvilli/filopodia, and that is helpful to understand how they interpret the results. I have no further concerns that have not been addressed in by the authors through new experiments or explanation.

RESPONSE TO REVIEWERS' COMMENTS

Response to Reviewer #1: (expertise in trogocytosis, extracellular vesicles):

Dear Reviewer,

We appreciate your comments regarding our revised manuscript entitled, “**Trogocytic-molting of T-cell microvilli controls T-cell clonal expansion (Reference: NCOMMS-22-23216A)**”. In response to your comments, my colleagues and I have made some changes to the revised manuscript, and we have addressed your comments, point by point. Our responses and accompanying revisions are as follows:

(General comments)

I commend the authors for the work they performed to revise the manuscript. The new experimentation has very much improved its scientific content and better support their conclusions. In particular, the experiments including LFA-1-deficient cells are well-designed and provided clear results. Also, the use of shorter time points and different substrates and adhesion molecules support the authors' statements. I only have a few remaining issues that need to be addressed:

Comment 1. The *in vivo* experiments demonstrate that TCRs are released from T cells, but uptake by DCs in these conditions is not addressed. These experiments do not need to be carried out for this already long story, but the conclusions regarding this part (lines 132-133) need to be toned down.

Answer → Thank you for your comments. Indeed, we performed the *in vivo* transfer of TCR β protein on the surface of cognate antigen-bearing DCs before (Kim et al, Nature Com 2018, 9:3630). We injected either OVA257-264 (MHC class I) or OVA323-339 (MHC class II) peptides into OTII TCR transgenic mice and evaluated the expression of TCR β^+ on the surface of DCs. Significant levels of TCR β were only observed on DCs from OVA323-339- but not OVA257-264-injected mice, demonstrating that TMPs were transferred into the cognate DCs *in vivo* (Please See Fig. 9c of Nature Com 2018, 9:3630).

Fig. 9 Physical interaction between TMPs and the DC surface is critical for TMP-mediated DC activation. **a** Schematic diagram of the transwell system (left). After co-incubation for 2 h, DCs were acid washed and evaluated for TCR β and V5G transfer (upper panel), followed continued incubation for 24 h. The expression of MHC class II, CD80, CD86, and CD40 on DCs (CD11c $^+$) was analysed by flow cytometry and expressed as histograms and mean fluorescence intensity (MFI). Data represent the mean of three experiments \pm SEM. * $P < 0.01$ vs. **c**. **b** 4-TMPs from anti-CD3-coated surfaces were treated with DCs as depicted in the schematic diagram (left) and incubated as described in **a**. TCR β , CD86, and CD40 presentation on DCs (CD11c $^+$) was analysed as described in **a**. Data represent the mean of three experiments \pm SEM. * $P < 0.01$ vs. **c**. **c** Transfer of TCRs to DCs *in vivo*. OTII TCR transgenic mice were administered OVA257-264 and OVA323-339 (100 μ g; i.p.) in PBS, respectively. After 48 h, draining lymph nodes were taken and dissociated, and cells were treated by acid buffer and stained for the indicated surface markers. Boxed areas represent TCR β expression on DCs. Data are represented by one of three independent experiments with similar results. * $P < 0.01$ vs. none or OVA257-264

Since there is already in vivo evidence of TCR transfer on the surface of cognate DCs, we have added a reference to support this information at the end of the sentence on line 133.

2. It is surprising that the authors observe no cytoplasmic presence of TCRzeta-GFP (Lines 140-142). This is somewhat different from what is seen with the endogenous protein. Please, specify and comment.

Answer → Thank you for your comments. As noted, upon expression of GFP-fused TCR ζ in T cells, we observed that the TCR ζ -GFP primarily localized on the cell membrane and microvilli, with some presence in the cytoplasm. We apologize for any confusion caused by the wording of the original statement. The word “exclusively” has been changed to “predominantly” in line 141. In addition, we added the words “with some presence in the cytoplasm” in line 142.

3. Please, quantify the results obtained by Western blot (densitometry and point-by-point representation including the number of replicates) throughout the manuscript.

Answer → Thank you for your comments. The western blot data in Fig. 4a was quantified and we have added the graph of densitometric analysis in Supplementary Fig. S8. Supplementary Fig. 9a was also quantitated.

4. The OCR/ECAR ratio does not add any new information, it is not mentioned by authors and can be misleading to the readers. Authors should remove it (Fig. 4). Are there differences in OCR and ECAR between NT, sAb and iAb? Please, include stats in the figure.

Answer → Thank you for your comments. We have agreed with your opinion that OCR/ECAR ratio is unnecessary figure, so we have removed it. Furthermore, we have conducted new experiments and replaced the results of Fig. 4h to make the data more clear."

5. Are the cells activated with iAb consuming less lipids than sAb? Authors should add an OCR/ECAR dataset using lipids as energy fuel to confirm this important issue.

Answer → Thank you for your critical comments. As previously shown in Figure 10, sAb-activated T cells have a greater preference for fatty acid oxidation (FAO) compared to iAb-activated T cells. Exogenous fatty acids are taken up by cells, converted to fatty acyl CoA in the cytosol, and then to acetyl CoA in the mitochondria through β -oxidation, eventually breaking down into CO_2 and H_2O . The process of converting fatty acids into CO_2 and H_2O through oxidation is referred to as fatty acid oxidation (FAO). To confirm this preference, we measured the fatty acid oxidation rate using palmitate-BSA and the levels of three different enzymes involved in β -oxidation: medium-chain acyl-CoA dehydrogenase (ACADM), very long chain acyl-CoA dehydrogenase (ACADVL), and trifunctional multi-enzyme complex subunit alpha (HADHA). As expected, sAb-activated T cells showed a significant immediate increase in OCR and high levels of FAO enzymes, indicating that they consume more lipids as energy fuel, compared to iAb-activated T cells. New experimental data has been added to Figs. 10f and 10g and additional description has been provided in lines 403-411.

Response to Reviewer #2: (expertise in immune synapse, immune cell signaling):

Dear Reviewer,

We appreciate your comments regarding our revised manuscript entitled, “**Trogoctytic-molting of T-cell microvilli controls T-cell clonal expansion (Reference: NCOMMS-22-23216A)**”. Your critical comments have made our claim clearer and more concise. Thank you again.

(General comments)

The authors addressed satisfactorily all the issues raised in my previous review with substantial experimental work.

I have a minor point that I noticed only on reading the revised manuscript to find the revised parts matching my critiques. Commenting on figure 4a, the authors state that there are no differences in signaling elicited by sAb versus iAb. This is not totally true in this figure (but it is clear in figure S8, which however does not include in the legend the sAb/iAb concentration used or the time of stimulation). For example, the phosphorylation of PKC θ appears more robust and sustained in response to sAb (10 microgram/ml condition). At the same Ab concentration pErk and pPLC γ appear lower in cells stimulated with sAb, but this might not be the case if the signals are normalized to the actin loading control, which is not homogeneous across the samples and is lowest at the 30 min time point of the 10 microgram/ml sAb sample. To clarify this point the immunoblots should be quantified over multiple experiments and quantifications shown (with stats).

Answer and revision → Thank you for your critical comments. According to your comments, we have conducted additional experiments to determine the exact signaling profile between sAb and iAb. The Western blot data was further quantified to address the reviewer's concerns. Upon normalizing the signals to the actin loading control, we found that there were no significant differences in signaling between sAb and iAb. Graphs of the densitometric analysis can be found in supplementary Figs. 8 and 9a, and the antibody concentrations used are listed in the legend of supplementary Fig. 9.

Response to Reviewer #3: (expertise in T cell activation, TCR exocytosis, immune synapse):

Dear Reviewer,

We appreciate your comments regarding our revised manuscript entitled, “**Trogocytic-molting of T-cell microvilli controls T-cell clonal expansion (Reference: NCOMMS-22-23216A)**”. Your critical comments have made our claim clearer and more concise. Thank you again.

Comment 1) The authors have responded adequately to my limited comments from the earlier review. They have performed ELISA type assays to look for deposition of sAb to well surfaces, leading to low density of iAb. I would have preferred if they use T cells to detect functional levels of iAb follow sAb based assays because T cells are more sensitive, but the Elisa is a standard approach. I was not aware that latrunculin A leads to shedding of microvilli/filopodia, and that is helpful to understand how they interpret the results. I have no further concerns that have not been addressed in by the authors through new experiments or explanation.

Answer and revision → We appreciate your overall comments on our revised manuscript again.

REVIEWERS' COMMENTS

Reviewer #1 (expert in trogocytosis and extracellular vesicles):

Authors have properly addressed the remaining comments of my review. The manuscript deserves publication in Nat Commun.